# Tokenizing 3D Molecule Structure with Quantized Spherical Coordinates

## Abstract

The application of language models (LMs) to molecular structure generation using line notations such as SMILES and SELFIES has been well-established in the field of cheminformatics. However, extending these models to generate 3D molecular structures presents significant challenges. Two primary obstacles emerge: (1) the difficulty in designing a 3D line notation that ensures SE(3)-invariant atomic coordinates, and (2) the non-trivial task of tokenizing continuous coordinates for use in LMs, which inherently require discrete inputs. To address these challenges, we propose Mol-StrucTok, a novel method for tokenizing 3D molecular structures. Our approach comprises two key innovations: (1) We design a line notation for 3D molecules by extracting local atomic coordinates in a spherical coordinate system. This notation builds upon existing 2D line notations and remains agnostic to their specific forms, ensuring compatibility with various molecular representation schemes. (2) We employ a Vector Quantized Variational Autoencoder (VQ-VAE) to tokenize these coordinates, treating them as generation descriptors. To further enhance the representation, we incorporate neighborhood bond lengths and bond angles as understanding descriptors. Leveraging this tokenization framework, we train a GPT-2 style model for 3D molecular generation tasks. Results demonstrate strong performance with significantly faster generation speeds and competitive chemical stability compared to previous methods. Further, by integrating our learned discrete representations into Graphormer model for property prediction on QM9 dataset, Mol-StrucTok reveals consistent improvements across various molecular properties, underscoring the versatility and robustness of our approach.

## 1 Introduction

The utilization of language models (LMs) for molecular generation has gained significant traction owing to their demonstrated success across various tasks (Irwin et al., 2022; Frey et al., 2023; Livne et al., 2024). Typically, molecules are represented as one-dimensional (1D) text strings using different line notation methods, such as SMILES (Weininger, 1988) and SELFIES (Krenn et al., 2020). These notations provide compact and discrete representations, which are well-suited to the discrete input requirements of language models. However, representing three-dimensional (3D) molecular structures

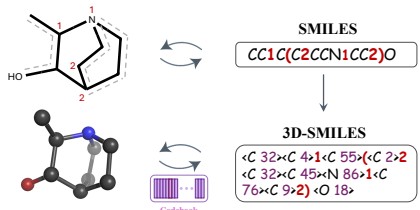

Figure 1: Converting continuous molecular structures into discrete token sequences.

presents challenges, and current approaches predominantly rely on graph-based representations (Luo et al., 2022; Hoogeboom et al., 2022). Graph-based methods are preferred because they effectively capture the intricate spatial relationships between atoms, which are crucial for accurately modeling 3D molecular structures.

Extending LMs to accommodate 3D molecular structures presents two primary challenges. First, there is a difficulty of sequentially defining SE(3)-invariant atomic coordinates based on line notations. SE(3) invariance ensures that the molecular structure remains consistent under translation and rotation. Second, the continuous nature of atomic coordinates is inherently incompatible with LMs, which are

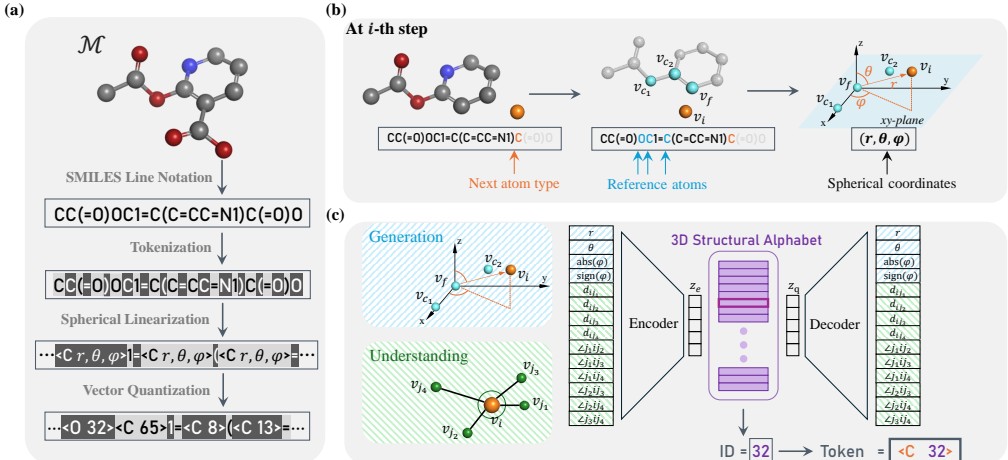

Figure 2: (a) Overview of the Mol-StrucTok pipeline. Taking SMILES notation as an example, the process begins with tokenizing the SMILES string to extract atom tokens. Each atom token is subsequently appended with its corresponding spherical coordinates. The coordinate extraction process for the $i$-th step is illustrated in (b). In (c), additional understanding descriptors are introduced, and a quantizer discretizes the concatenated continuous descriptors. For clarity, hydrogen atoms (H) are omitted.

designed to process discrete inputs. Previous works have attempted to tokenize continuous scalar values into discrete digits for coordinate modeling (Zholus et al., 2024; Li et al., 2024), but such tokenization methods often struggle with capturing numerical semantics and generalizing to unseen values (Golkar et al., 2023).

To address these challenges, we propose a novel method for 3D molecular structure tokenization, called Mol-StrucTok. Our approach involves constructing a spherical line notation and subsequently discretizing the continuous atomic coordinates. In the first step, we develop a spherical line notation tailored to 3D molecular structures. Similar to autoregressive models in 3D molecule generation (Gebauer et al., 2019; Simm et al., 2020; Daigavane et al., 2023), we sequentially place atoms in 3D space, defining their distances, bond angles, and torsion angles through a local spherical coordinate system. However, we diverge by constructing this spherical coordinate system independently, without relying on a predictive model to determine reference points. By appending each atom's local spherical coordinates to its corresponding token in the original line notation, we preserve both the molecular graph topology and the SE(3)-invariant 3D structural information.

In the second step, we utilize a vector quantized variational autoencoder (Van Den Oord et al., 2017) to discretize the continuous coordinates, as continuous vectors must be transformed for autoregressive language modeling approaches (Vaswani, 2017; Radford et al., 2019; Ramesh et al., 2021). Drawing inspiration from the work of van Kempen et al. (2022), we treat each atom as a data point and use its spherical coordinates as generation descriptors. Additionally, to enrich the representation of the atomic environment, we introduce supplementary descriptors, including bond lengths and bond angles of neighboring atoms, which provide a more comprehensive understanding of the local atomic structure.

Building on this framework, we train a GPT-2 model (Radford et al., 2019) and achieve strong performance in 3D molecular generation tasks. Prior autoregressive models (Gebauer et al., 2019; Simm et al., 2020; Daigavane et al., 2023) often resulted in molecules with low chemical stability, while diffusion models (Hoogeboom et al., 2022; Xu et al., 2023; You et al., 2023), though more effective, incurred high computational costs due to the extensive number of diffusion steps. Our method strikes a balance by delivering competitive chemical stability while significantly reducing generation time. Additionally, the powerful conditional generation capabilities of language models enable our approach to excel in tasks demanding conditional molecular generation, yielding substantial improvements in both efficiency and accuracy.

Furthermore, we show that the learned discrete representations are also beneficial for molecular understanding tasks. By incorporating the learned structure tokens as additional embeddings in

the Graphormer model (Ying et al., 2021), we observe consistent performance enhancements in property prediction tasks, as evidenced by results on the QM9 dataset. This underscores the broader applicability and effectiveness of our Mol-StrucTok framework across generation and prediction tasks.

## 2 RELATED WORK

**3D Molecule Generation.** Early works such as G-Schnet (Gebauer et al., 2019), MolGym (Simm et al., 2020), and G-SphereNet Luo & Ji (2022) focus on autoregressively predicting atom types and their connectivity in sequence. Alternative generative models, such as Variational Autoencoders (VAE) (Nesterov et al., 2020) and Normalizing Flow models (Garcia Satorras et al., 2021), have explored single-shot generation strategies. With the rise of diffusion models, numerous studies have demonstrated their remarkable strength in 3D molecular modeling (Hoogeboom et al., 2022; Xu et al., 2023; You et al., 2023; Song et al., 2023), consistently outperforming autoregressive counterparts. However, recent autoregressive approaches, such as Symphony (Daigavane et al., 2023) and Geo2Seq (Li et al., 2024), have also shown promising results, bridging the performance gap and suggesting continued relevance of this methodology in molecular generation.

**Structure Tokenization.** Recent advancements in tokenizing 3D structures for biomolecules, particularly proteins, have gained considerable attention. FoldSeek (van Kempen et al., 2022) first used VQ-VAE to tokenize protein structures for improving structure alignment. Building on this, researchers applied this structure alphabet to models like ESM (Su et al., 2023) and T5 (Heinzinger et al., 2023), providing discrete structural embeddings that aid in protein understanding. ESM3 (Hayes et al., 2024) and FoldToken (Gao et al., 2024) have further explored the use of VQ-VAE's discrete representations to predict protein structures. However, similar approaches have not been developed for small molecules, as their lack of a well-defined sequence makes tokenization into ordered representations more challenging. Some studies have attempted to tokenize the continuous coordinates of small molecules into digits (Zholus et al., 2024; Li et al., 2024), but this approach renders models highly sensitive to numerical length, impairs their ability to capture semantic relationships between similar values, and hinders generalization to unseen data.

## 3 METHODS

Our proposed Mol-StrucTok consists of two key components: constructing a spherical line notation and discretizing continuous atomic coordinates. This approach aims to preserve both molecular topology and SE(3)-invariant structural information.

### 3.1 PRELIMINARY: AUTOREGRESSIVE GENERATION OF 3D MOLECULES

**Molecular Notation**. A molecule is modeled as a graph $\mathcal{G} = (\mathcal{V}, \mathcal{E})$, where $\mathcal{V}$ represents the set of atoms, and $\mathcal{E}$ represents the set of chemical bonds between them. Additionally, we use $\mathcal{R}$ to denote the conformation of $\mathcal{G}$, forming a complete 3D molecular structure $\mathcal{M} = (\mathcal{G}, \mathcal{R}) = (\mathcal{V}, \mathcal{E}, \mathcal{R})$. Each atom $v_i \in \mathcal{V}$ is associated with a 3D spatial coordinate $\boldsymbol{x}_i \in \mathbb{R}^3$, and its atom type is represented by $z_i \in \mathbb{Z}$, where $z_i$ is the atomic number (nuclear charge) of the $i$-th atom.

In earlier tasks of 2D molecular graph generation, many works have modeled this problem in an autoregressive manner (Popova et al., 2019; Shi et al., 2020; Luo et al., 2021). They iteratively predicting new atom types and bond formations until no new atoms can be added or no bonds can be formed with the current molecule. Similarly, autoregressive generation of 3D molecules follows a process where new atoms are sequentially placed in 3D space (Simm et al., 2020; Luo & Ji, 2022; Zhang et al., 2023). At the $i$-th step, denote the intermediate 3D molecular geometry generated from the preceding $i - 1$ steps as $\mathcal{M}_i = (\mathcal{G}_i, \mathcal{R}_i)$, consisting of $i$ atoms. This process can be described as constructing a sequence of increasingly larger structures: $\mathcal{M} = \{\mathcal{M}_0, \mathcal{M}_1, \ldots, \mathcal{M}_{|\mathcal{V}|}\}$, where at step $i$, the $i$-th atom type $z_i$ and its spatial coordinates $\boldsymbol{x}_i$ are generated. A local coordinate system is established at each stage $\mathcal{M}_i$, predicting the atom's local coordinates, which are then transformed into global spatial coordinates. The process is divided into three main steps:

1. *Predict next atom type* based on the partial molecular structure.

2. *Establish a local spherical coordinate system.* A coordinate system is defined by selecting three reference atoms. Typically, a focal atom $f$ is chosen, along with its closest and second-closest atoms, $c_1$ and $c_2$, respectively [1].

3. *Define the new atom coordinates.* In the spherical coordinate system, the model predicts the descriptors $d_i$, $\theta_i$, and $\varphi_i$. Here, $d_i \in \mathbb{R}^+$ is the distance between the new atom and $f$, while $\theta_i \in [0, \pi]$ and $\varphi_i \in (-\pi, \pi]$ are angular coordinates. Some later works, like Daigavane et al. (2023), avoid spherical coordinates in favor of orientation prediction to better handle symmetry.

The order of the first and second steps can vary in some works (Simm et al., 2020). Each step involves specific model predictions, described as follows:

$$z_i = f_z(\mathcal{M}_{i-1}) \quad f = f_p(\mathcal{M}_{i-1}, z_i)$$
$$d_i = f_d(\mathcal{M}_{i-1}, p_f, z_i) \quad \theta_i = f_\theta(\mathcal{M}_{i-1}, p_f, z_i) \quad \varphi_i = f_\varphi(\mathcal{M}_{i-1}, p_f, z_i). \tag{1}$$

In this work, we iteratively construct a spherical coordinate system to develop a spherical line notation, without introducing any parameterized models—marking a key distinction from prior approaches.

## 3.2 3D Spherical Line Notation

**A Generalized Line Notation.** To represent a molecule in a linear format, we tokenize the molecular graph into a sequence of atom tokens $\mathbb{A}$ and non-atom tokens $\mathbb{B}$. Let $\mathbb{T} = \mathbb{A} \cup \mathbb{B}$ be the set of all possible tokens. The sequence is constructed as follows:

- *Atom Tokens.* Each atom $v_i \in \mathcal{V}$ is mapped to an atom token $t_i \in \mathbb{A}$. These tokens typically correspond to the atomic type of the atoms in the molecule [2].

- *Non-Atom Tokens.* Non-atom tokens correspond to chemical bonds between atoms or other structural features such as branching or ring closure. Each bond $e_{ij} \in \mathcal{E}$ is mapped to a non-atom token $t_{ij} \in \mathbb{B}$, indicating the bond type (e.g., single, double, or triple bond).

Thus, for a molecule $\mathcal{M}$, the tokenized sequence $s$ can be expressed as:

$$s = (t_1, t_2, \ldots, t_m), \tag{2}$$

where each $t_i$ is either an atom token $t_i \in \mathbb{A}$ or a non-atom token $t_i \in \mathbb{B}$, $m$ is the total number of tokens in the sequence.

**Augmented Atom Tokens with Spherical Features.** As demonstrated in Section 3.1, by appropriately constructing the spherical coordinate system in a sequential manner, the complete structure of a molecule can be accurately reconstructed. The local spherical coordinates of each atom can themselves form a sequence when arranged in a specific order. In this context, we can adopt the sequence of the line notation $s$ as the ordering scheme.

For each atom token $t_i \in \mathbb{A}$, in addition to being labeled by its atomic type, will also carry its position in 3D space. The position of each atom is described using spherical coordinates: $(d_i, \theta_i, \varphi_i)$ as described in Section 3.1. Thus, the augmented atom token for each atom token $t_i$ becomes:

$$r_i = (t_i, d_i, \theta_i, \varphi_i). \tag{3}$$

**Establish the Spherical Coordinate System.** The only remaining question is how to construct the spherical coordinate system for each atom. Previous autoregressive methods rely on models to predict a focal atom and select two reference atoms based on spatial Euclidean distance. However, since our approach is model-free, we explore three alternatives for selecting reference atoms when establishing the local coordinate system, as shown in Figure 3. When constructing the coordinate system for atom $v_i$, we consider the following approaches to determine the indices of the focal atom and two references, denoted as $f, c_1, c_2$:

- *1D Sequence-Based Reference Selection.* In this approach, we select the focal atom and reference atoms based on their sequential order in the line notation. The preceding atoms in the sequence are chosen as the reference points:

$$f = i - 1, \quad c_1 = i - 2, \quad c_2 = i - 3. \tag{4}$$

---

[1] Different approaches may vary slightly; for instance, in G-SphereNet (Luo & Ji, 2022), $c_1$ is selected as the atom closest to $f$, and $c_2$ is the atom closest to $c_1$.

[2] In SELFIES (Krenn et al., 2020), the atom tokens may also contain bond information, such as "[=C]".

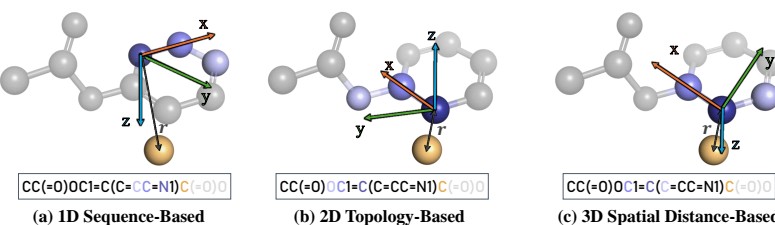

| (a) 1D Sequence-Based | (b) 2D Topology-Based | (c) 3D Spatial Distance-Based |

Figure 3: Three types of reference atom selection.

However, this method poses a significant issue: sequentially adjacent atoms may be spatially distant, resulting in descriptor outliers, which degrade the model's predictive performance. Additionally, atoms that are adjacent in the molecular topology but distant in the sequence can lead to accumulated errors in the autoregressive generation, causing bond lengths between connected atoms to become overly stretched or compressed.

- *2D Topology-Based Reference Selection.* To mitigate the aforementioned issues, we propose selecting the focal atom as the closest atom in sequence that is also topologically connected. This ensures that the focal atom and reference atoms are spatially close, reducing the likelihood of prediction outliers. The reference points are selected as follows:

$$
\begin{aligned}
f = F(i) = \mathrm{argmax}_j(\mathcal{N}(i,j)), & \quad \text{s.t. } j < i, \\
c_1 = F(f), \quad c_2 = F(c_1).
\end{aligned}
\tag{5}
$$

where $\mathcal{N}(i,j)$ denotes the topological neighbors of atom $v_i$ with respect to atom $v_j$. This method ensures that $v_i$, $v_f$, $v_{c_1}$ and $v_{c_2}$ are adjacent in the molecular topology, reducing the likelihood of generating distant, unpredictable atom placements.

- *3D Spatial Distance-Based Reference Selection.* Inspired by previous works (Simm et al., 2020; Luo & Ji, 2022; Zhang et al., 2023), we further explore an alternative approach in which reference atoms are selected based on their spatial distance to the focal atom. After selecting the focal atom topologically, the closest reference atoms are determined using Euclidean distance. The reference selection process is formalized as:

$$
\begin{aligned}
f &= \mathrm{argmax}_j(\mathcal{N}(i,j)), & \text{s.t. } j < i, \\
c_1 &= \mathrm{argmax}_k \|\boldsymbol{x}_f - \boldsymbol{x}_k\|, & \text{s.t. } k \neq f \text{ and } k < i, \\
c_2 &= \mathrm{argmax}_k \|\boldsymbol{x}_f - \boldsymbol{x}_k\|, & \text{s.t. } k \neq f, k \neq c_1, \text{ and } k < i,
\end{aligned}
\tag{6}
$$

where $\|\cdot\|$ represents the Euclidean distance. Although this method leverages spatial proximity for reference selection, our analysis in Section 5 reveals that when multiple atoms are equidistant from the focal atom, small coordinate perturbations can lead to incorrect selections of $c_1$ and $c_2$. This, in turn, may cause the reconstructed molecular structure to collapse in the wrong direction, leading to significant distortions.

We ultimately adopt the 2D topology-based selection, as it balances topological and spatial proximity, reducing prediction errors. Further analysis can be found in Section 5.

### 3.3 DISCRETIZATION WITH STRUCTURAL ALPHABET

Spherical coordinates are inherently continuous vectors. To apply autoregressive modeling for next-token prediction, these vectors must be quantized into discrete tokens through vector quantization techniques. VQ-VAE (Van Den Oord et al., 2017), as a widely used method, transforms an image into a set of discrete codes within a learnable latent space. We adopt this approach for our task.

**Descriptors for 3D Structural Alphabet.** For $i$-th atom in a molecular structure, we have its local spherical coordinates as $(d_i, \theta_i, \varphi_i)$. Since $\theta_i$ lies in $[0, \pi]$ and $\varphi$ in $(-\pi, \pi]$, to normalize their ranges, we decompose $\varphi_i$ into its sign and absolute value. Thus, the generation descriptor is defined as:

$$
\boldsymbol{g}_i = (d_i, \theta_i, \mathrm{abs}(\varphi_i), \mathrm{sign}(\varphi_i)).
\tag{7}
$$

In addition to the generation descriptors $\boldsymbol{g}_i$, we also consider understanding descriptors based on the local atomic environment. For each atom $v_i$, these include the bond lengths and pairwise bond angles with its four nearest neighbors, depicted in Figure 2. Let the understanding descriptors for atom $v_i$ be:

$$\boldsymbol{u}_i = (l_{j_1}, l_{j_2}, l_{j_3}, l_{j_4}, \alpha_{j_1 i j_2}, \alpha_{j_1 i j_3}, \alpha_{j_1 i j_4}, \alpha_{j_2 i j_3}, \alpha_{j_2 i j_4}, \alpha_{j_3 i j_4}), \tag{8}$$

where $l_{j_k}$ represents the bond length between atom $v_i$ and its $k$-th closest neighboring atom $v_{j_k}$, and $\alpha_{j_k i j_l}$ denotes the bond angle formed by the bond between $v_i$ and $v_{j_k}$ and the bond between $v_i$ and $v_{j_l}$, for $1 \leq k < l \leq 4$.

We concatenate the generation and understanding descriptors to form a combined descriptor $\boldsymbol{z}_i$ for each atom:

$$\boldsymbol{z}_i = [\boldsymbol{x}_i; \boldsymbol{u}_i] = [d_i, \theta_i, \text{abs}(\varphi_i), \text{sign}(\varphi_i), l_{j_1}, l_{j_2}, l_{j_3}, l_{j_4}, \alpha_{j_1 i j_2}, \alpha_{j_1 i j_3}, \alpha_{j_1 i j_4}, \alpha_{j_2 i j_3}, \alpha_{j_2 i j_4}, \alpha_{j_3 i j_4}]. \tag{9}$$

This concatenated descriptor $\boldsymbol{z}_i \in \mathbb{R}^{14}$ contains the 3D spatial information and the local atomic environment for atom $v_i$. The SE(3)-invariance of the descriptors is proven in the Appendix B.

**Vector Quantization.** Each descriptor $\boldsymbol{z}_i$ is first encoded by an encoder $\mathcal{E}$, yielding an embedding $f = \mathcal{E}(\boldsymbol{z}_i)$, which is then mapped by a quantizer $\mathcal{Q}$ to a discrete token $q_i$. The quantizer typically includes with a learnable codebook $\mathcal{C} = \{\boldsymbol{c}_i\}_{i=1}^{K}$, containing $K$ vectors. During quantization, the feature vector is mapped to the closest code in the codebook based on the nearest code index $q_i$, as defined by:

$$q_i = \arg\min ||\mathcal{E}(\boldsymbol{z}_i) - \boldsymbol{c}_j||. \tag{10}$$

Intuitively, $\boldsymbol{c}_i$ serves as the estimation of $f$. The $\boldsymbol{c}_i$ is subsequently passed through a decoder $\mathcal{D}$ to generate the reconstructed descriptor. To train this quantized autoencoder, we aim to achieve both accurate reconstruction and precise estimation of $f$ by $\boldsymbol{c}_i$. The overall objective is described in Equation 11. The operator sg represents a stop-gradient operation, which prevents gradients from being backpropagated through its argument (Van Den Oord et al., 2017), and $\beta$ is a hyperparametr that regulates the resistance to modifying the code corresponding to the encoder's output.

$$\mathcal{L}_{\text{Tokenizer}} = ||\boldsymbol{z}_i - \mathcal{D}(\boldsymbol{c}_i)||_2^2 + ||\text{sg}[\mathcal{E}(\boldsymbol{z}_i) - \boldsymbol{c}_i]||_2^2 + \beta||\text{sg}[\boldsymbol{c}_i] - \mathcal{E}(\boldsymbol{z}_i)||_2^2. \tag{11}$$

Additionally, We replace the codebook loss (the second loss term in Equation 11) with updates using an exponential moving average for the codebook (Razavi et al., 2019). Thus, each continuous descriptor $\boldsymbol{z}_i$ is assigned to a discrete token $c_k$ from the codebook.

## 3.4 AUTOREGRESSIVE MODELING WITH GPT-2

**Expanding Vocabulary.** As describe in Section 3.2, expanding the atom token set $\mathbb{A}$ involves generating the Cartesian product of the structural alphabet set $\mathbb{S}$ and the atom type set $\mathbb{A}$.

$$\mathbb{A}_{\text{expand}} = \mathbb{A} \times \mathbb{S} = \{(a_i, s_j) | a_i \in \mathbb{A}, s_j \in \mathbb{S}\}. \tag{12}$$

For instance, the atom type C combined with its corresponding structural alphabet 32 forms the token $\langle$C 32$\rangle$. This concept, originating from the work of Su et al. (2023), aims to establish a structure-aware vocabulary that integrates both structural and atomic identifiers. For the non-atom token set $\mathbb{B}$, we assign a structural token value of $-1$ to simplify its representation.

$$\mathbb{B}_{\text{expand}} = \{(b_i, -1) | b_i \in \mathbb{B}\}. \tag{13}$$

The final vocabulary is then constructed as:

$$\mathbb{V} = \mathbb{A}_{\text{expand}} \cup \mathbb{B}_{\text{expand}} = \{(a_i, s_j) | a_i \in \mathbb{A}, s_j \in \mathbb{S}\} \cup \{(b_i, -1) | b_i \in \mathbb{B}\}. \tag{14}$$

**Autoregressive Modeling.** Based on the defined sequence $s = (t_1, t_2, \ldots, t_m)$ for $t_i \in \mathbb{V}$, we can model the molecular generation process using an autoregressive language model. Specifically, we use GPT-2 (Radford et al., 2019), a powerful and widely available model, to verify the capacity of this approach. The objective is to minimize the expected prediction error over the entire sequence:

$$\mathcal{L}_{LM} = -\sum_{i=1}^{m} \log p(t_i | t_1, t_2, \ldots, t_{i-1}). \tag{15}$$

**Controllable Generation.** To enable controllable generation, we introduce a condition scalar, which is tokenized into a sequence of digits. For example, for condition $c = -1.34$: tok$(-1.34) = (-, 1, ., 3, 4)$. This condition is prepended to the sequence $s$ to form $s' = (\text{tok}(c), t_1, t_2, \ldots, t_m)$, allowing the model to generate molecules based on the provided condition.

Table 1: Validity and uniqueness (among valid) percentages of molecules with different bond assignment methods, with **best** and second-best models highlighted. Results of * are obtained by our experiments. Other results are borrowed from Daigavane et al. (2023).

| Metric ↑ | QM9 | Symphony | EDM | G-SchNet | G-SphereNet | GeoLDM* | LDM-3DG* | Mol-StrucTok |
|---|---|---|---|---|---|---|---|---|
| Validity via `xyz2mol` | 99.99 | 83.50 | 86.74 | 74.97 | 26.92 | 91.32 | **98.68** | 96.67 |
| Validity via OpenBabel | 94.60 | 74.69 | 77.75 | 61.83 | 9.86 | 88.85 | 91.64 | **92.16** |
| Validity via Lookup Table | 97.60 | 68.11 | 90.77 | 80.13 | 16.36 | 93.76 | 94.89 | **98.02** |
| Uniqueness via `xyz2mol` | 99.84 | 97.98 | **99.16** | 96.73 | 21.69 | 98.85 | 98.22 | 85.35 |
| Uniqueness via OpenBabel | 99.97 | 99.61 | **99.95** | 98.71 | 7.51 | 99.93 | 98.13 | 84.71 |
| Uniqueness via Lookup Table | 99.89 | 97.68 | **98.64** | 93.20 | 23.29 | 98.18 | 97.03 | 85.11 |
| Atom Stability | 99.0 | - | 98.7 | 95.7 | - | **98.91** | 97.57 | 98.54 |
| Molecule Stability | 95.2 | - | 82.0 | 68.1 | - | **89.79** | 86.87 | 88.30 |

## 4    EXPERIMENTAL RESULTS

We evaluated Mol-StrucTok across several aspects to assess its ability to capture valid 3D structure distributions, demonstrate strong conditional generation performance, and provide informative structural tokens that serve as robust 3D representations.

### 4.1    SETUP

**Vector Quantization.** To achieve atom-level quantization, we trained a vector quantized variational autoencoder on the PCQM4Mv2 dataset (Nakata & Shimazaki, 2017), which contains 3.4 million organic molecular structures and approximately 99 million atomic descriptors. We opted for a lightweight architecture, inspired by the simplicity of the Foldseek (van Kempen et al., 2022). Specifically, we implemented a 3-layer MLP encoder and a 2-layer MLP decoder, totaling 74k parameters. The hidden dimension was set to 128, and the latent space embedding dimension was chosen as 5. The structural vocabulary size was defined as 256. A larger vocabulary leads to lower reconstruction error, but presents greater challenges in downstream tasks. Further configurations of the quantizer can be found in the Appendix Section C.1.

**Dataset.** Leveraging the learned structural alphabet, we conducted comprehensive evaluations on downstream tasks, primarily focusing on the QM9 dataset (Ramakrishnan et al., 2014). QM9 is a widely used quantum chemistry dataset that provides one equilibrium conformation and 12 geometric, energetic, electronic, and thermodynamic properties for 134,000 stable organic molecules composed of CHONF atoms. If not stated, the data split follows standard settings, with 110,000 samples for training, 10,000 for validation, and the remaining 10,831 samples used for testing.

**Generation.** For the molecular generation tasks, we adopted the GPT-2 (Radford et al., 2019) architecture with 12 layers and a hidden dimension of 768. Using the SELFIES line notation, we generated a vocabulary of 2,331 tokens [3]. Mol-StrucTok was evaluated in both unconditional and conditional generation settings to assess its ability to generate valid molecular structures. Unless explicitly stated otherwise, the decoding strategy utilizes multinomial sampling with a top-k value of 50 and a temperature setting of $\tau = 0.7$. The implementation details are provided in the Appendix.

**Understanding.** In the molecular understanding tasks, we used Graphormer (Ying et al., 2021) as the base model, with 12 layers and a hidden size of 768. No modifications were made to the model architecture aside from augmenting the atom-type embedding with the Mol-StrucTok embedding. We then evaluated its performance on the QM9 property prediction tasks to assess its effectiveness in capturing molecular properties. The detailed model configurations are provided in the Appendix C.3.

### 4.2    UNCONDITIONAL GENERATION

Although our method generates both topology and structure [4], similar to LDM-3DG (You et al., 2023) but unlike other previous works, we can still treat the generated structure as a point cloud and apply bond order assignment tests. These tests ensure that the atomic distances are physically

---

[3]atomic types × structural vocabulary + non-atom token types.

[4]This capability is crucial for downstream tasks such as molecular editing and reaction prediction.

Table 2: Percentage of valid (as obtained from `xyz2mol`) molecules passing each PoseBusters test. Results of * are obtained by our experiments. Other results are borrowed from Daigavane et al. (2023). We highlight **best** and second-best results.

| Test ↑ | Symphony | EDM | G-SchNet | G-SphereNet | GeoLDM* | LDM-3DG* | MOL-STRUCTOK |
|---|---|---|---|---|---|---|---|
| All Atoms Connected | 99.92 | 99.88 | 99.87 | **100.00** | 99.95 | **100.00** | **100.00** |
| Reasonable Bond Angles | 99.56 | 99.98 | 99.88 | 97.59 | 99.97 | 99.83 | **100.00** |
| Reasonable Bond Lengths | 98.72 | **100.00** | 99.93 | 72.99 | 99.96 | 99.89 | 99.99 |
| Aromatic Ring Flatness | **100.00** | **100.00** | 99.95 | 99.85 | **100.00** | 99.99 | 99.91 |
| Double Bond Flatness | 99.07 | 98.58 | 97.96 | 95.99 | 99.20 | 99.35 | **99.87** |
| Reasonable Internal Energy | 95.65 | 94.88 | 95.04 | 36.07 | 96.89 | 97.75 | **97.77** |
| No Internal Steric Clash | 98.16 | 99.79 | 99.57 | 98.07 | 99.65 | 99.94 | **99.97** |

Table 3: Conditional generation on six quantum properties evaluation. A lower number indicates a better controllable generation result.

| Property | $\alpha$ | $\Delta\varepsilon$ | $\varepsilon_{\text{HOMO}}$ | $\varepsilon_{\text{LUMO}}$ | $\mu$ | $C_v$ |
|---|---|---|---|---|---|---|
| Units | Bohr$^3$ | meV | meV | meV | D | $\frac{\text{cal}}{\text{mol}}$K |
| QM9 | 0.10 | 64 | 39 | 36 | 0.043 | 0.040 |
| Random | 9.01 | 1470 | 645 | 1457 | 1.616 | 6.857 |
| EDM | 2.76 | 655 | 356 | 584 | 1.111 | 1.101 |
| GeoLDM | 2.37 | 587 | 340 | 522 | 1.108 | 1.025 |
| GeoBFN | 2.34 | 577 | 328 | 516 | 0.998 | 0.949 |
| **MOL-STRUCTOK** | **0.33** | **89** | **64** | **62** | **0.285** | **0.169** |

Table 4: Mean Absolute Error for QM9 property prediction.

| Property | $\varepsilon_{\text{HOMO}}$ | $\varepsilon_{\text{LUMO}}$ | $\Delta\varepsilon$ |
|---|---|---|---|
| Units | meV | meV | meV |
| SchNet | 41 | 34 | 63 |
| NMP | 43 | 38 | 69 |
| EDM | 34 | 38 | 61 |
| Graphormer | 46 | 47 | 66 |
| + **MOL-STRUCTOK** | **42** | **39** | **62** |

reasonable. We follow Daigavane et al. (2023) to carry out bond assignment with multiple tools, including `xyz2mol` (Kim & Kim, 2015), OpenBabel (O'Boyle et al., 2011), and a simple lookup table (Hoogeboom et al., 2022). A generated molecule is considered valid if an algorithm in `xyz2mol` or OpenBabel successfully assigns bonds without causing charge imbalances. In the Lookup Table, validity means the molecule can be converted to SMILES strings. The additional stability metric ensures no charge imbalances occur via Loopup Table [5]. We also evaluate the uniqueness of valid molecules by calculating the proportion of duplicate structures in SMILES notation. The results of these evaluations, based on 10,000 generated molecules, are presented in Table 1.

Diffusion-based methods, such as EDM (Hoogeboom et al., 2022), GeoLDM (Xu et al., 2023), and LDM-3DG (You et al., 2023), have demonstrated superior performance in modeling molecular validity and uniqueness compared to autoregressive approaches like G-SchNet (Gebauer et al., 2019), G-SphereNet (Luo & Ji, 2022), and Symphony (Daigavane et al., 2023). Our method exhibits highly competitive results in terms of validity and stability. However, its performance in the uniqueness metric is relatively limited, as diffusion models utilize a stepwise process that introduces controlled noise, enabling greater variability in the generated outputs. Additionally, the decoding temperature $\tau$ has a significant impact on the trade-off between diversity and quality. A detailed analysis of this trade-off is provided in Section 5.

Beyond bond assignment, we employed the PoseBusters test suite (Buttenschoen et al., 2024), which provides multiple sanity checks for evaluating the validity of protein-ligand complexes. Although designed for complexes, we applied the small molecule-specific metrics to assess the validity of our generated molecules. As shown in Table 2, our method performs well across all evaluated criteria.

### 4.3 CONDITIONAL GENERATION

Apart from evaluating the validity of generated molecules, a more practical challenge is the ability to generate molecules conditioned on specific target properties. In this section, we focus on six critical quantum mechanical properties from the QM9 dataset: polarizability ($\alpha$), orbital energies ($\varepsilon_{\text{HOMO}}$, $\varepsilon_{\text{LUMO}}$), their energy gap ($\Delta\varepsilon$), dipole moment ($\mu$), and heat capacity ($C_v$). Following Hoogeboom et al. (2022), the generator is trained on half of the training set, where it generates 3D molecular structures conditioned on specific target properties. The remaining half is used to train a classifier that scored the quality of the generated molecules. To ensure fairness, we employed the EGNN classifier (Satorras et al., 2021) trained by Hoogeboom et al. (2022) and use the non-overlapping data

---

[5]This convoluted definition follows Symphony, as it conflates validity and stability in the lookup table.

Table 5: Fraction of RMSD<1Å reconstructions under specific noise scale for three types of reference frame selection.

| Noise Scale | 1D-based | 2D-based | 3D-based |
|---|---|---|---|
| 0.01 | 81.74 | 82.37 | 33.29 |
| 0.05 | 77.15 | 81.01 | 19.00 |
| 0.10 | 65.38 | 78.94 | 13.16 |

Table 6: Unconditional generation comparisons between SMILES and SELFIES.

| Metric ↑ | SMILES | SELFIES |
|---|---|---|
| Validity | 97.48 | 98.02 |
| Uniqueness | 81.52 | 85.11 |
| Atom Stability | 98.64 | 98.54 |
| Molecule Stability | 89.71 | 88.30 |

to train the generative model. The primary evaluation metric was the Mean Absolute Error (MAE) between the given property $\xi$ and the predicted property $\hat{\xi}$ of the generated molecule. A lower MAE reflects the model's ability to accurately generate molecules with the desired quantum properties.

Our results in Table 3 demonstrate significant improvements over the previous state-of-the-art method, GeoBFN (Song et al., 2023). The QM9 row represents the mean absolute error (MAE) of the classifier's prediction on ground truth properties, indicating oracle performance. We attribute this improvement to Mol-StrucTok's use of a language model, which offers greater determinism compared to diffusion models. This enhanced determinism allows for more accurate reliance on scalar properties. Additionally, by linearizing molecular structures through a sequential representation, the language model simplifies the complexity of the problem, contributing to the observed performance gains.

### 4.4 QM9 PROPERTY PREDICTION

Additionally, we investigate how the learned discrete representations from Mol-StrucTok enhance molecular understanding, such as predicting molecular properties. To assess the model's understanding capability, we integrated the learned structure tokens as additional embeddings into a 2D Graph-based model, Graphormer (Ying et al., 2021). The model was then evaluated on predicting three key properties from the QM9 dataset: $\epsilon_{\text{HOMO}}$, $\epsilon_{\text{LUMO}}$, and their gap $\delta\epsilon$. These properties are crucial for analyzing the electronic behavior of molecules.

The main baseline for this evaluation is the vanilla Graphormer without the Mol-StrucTok structural embeddings. We also report several baseline 3D molecular understanding models, including SchNet (Schütt et al., 2018) and NMP (Gilmer et al., 2017), for references. This task requires complex model designs, making it difficult to compare with state-of-the-art models. The primary evaluation metric used was the Mean Absolute Error (MAE) of the predicted molecular properties.

The results show that incorporating Mol-StrucTok embeddings enables the 2D-based Graphormer to reach the performance of baseline 3D models. This highlights the effectiveness of the learned discrete representations in bridging the gap between 2D and 3D molecular property prediction, confirming that the structure tokens provide valuable insights into molecular understanding tasks.

## 5 FURTHER ANALYSIS

**Robustness of Reference Node Selection.** To assess the robustness of the three reference frames defined in Section 3.2 under noise, we design the following evaluation: First, the ground truth coordinates are transformed into corresponding descriptors. Noise is then introduced at varying magnitudes (0.05, 0.01, 0.1) to the descriptors, after which they are reconstructed back into coordinates for comparison. We report the fraction of reconstructions with an RMSD of less than 1Å between the ground truth and reconstructed coordinates in Table 5. Our analysis reveals that the 3D spatial distance-based reference frame performs the worst under noisy conditions due to the ambiguous selection of $c_1$ and $c_2$. In contrast, the 2D topology-based reference frame shows the highest tolerance to noise, even outperforming the sequence-based approach. Notably, with noise scaled to 0.1, the 2D reference frame still achieves a 78.94% rate of reconstructions with RMSD < 1Å.

**Properties of Structural Alphabet.** We seek to explain some of the data distributions captured by our structural alphabet, such as bond length. Specifically, we decoded each alphabet, with the vocabulary size set 256, into its corresponding descriptor space, where bond length $d$ is the first element of the generation descriptor. We plotted the bond length distribution of the 256 tokens, also

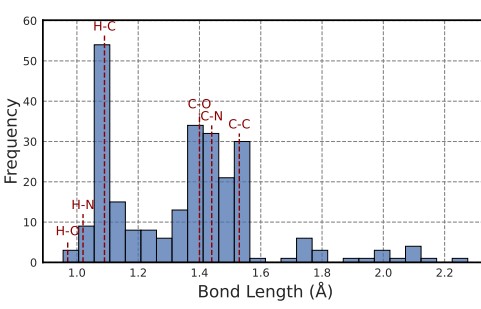 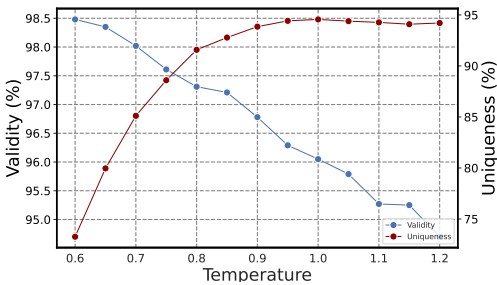

(a) Bond length distribution decoded from the alphabet.    (b) Influence of temperature $\tau$ in decoding strategy.

Figure 4: Analysis of structural alphabet and temperature.

highlighting the six most common bond types (calculated from all PCQM4Mv2 samples). The results demonstrate that our vocabulary provides broad coverage of bond lengths. Moreover, we observed that among the six bond types, the local structure containing H-O exhibited the least variability, whereas C-H structures were the most diverse. Additionally, C in combination with other heavy atoms formed a wide variety of structures.

**Line Notation Variants.** In our experiments, we used the SELFIES representation as it is better suited for language model. Here, we compared it with SMILES variants under the same settings. Both can produce syntax errors: SMILES may produce unbalanced brackets, incorrect ring closures, or valence violations, while SELFIES can generate meaningless atom tokens. For well-converged models, syntax errors occur in fewer than 10 out of 10,000 samples, making them negligible. Table 6 shows that the validity, uniqueness, and stability of error-free molecules are similar between the two notations, indicating that our spherical line notation is generalizable to any line notation.

**Analysis of Temperature in Decoding.** The temperature parameter $\tau$ in the decoding strategy significantly affects model performance, balancing diversity and quality. In Figure 4b, we plotted the curves of validity and uniqueness against varying temperature values in unconditional generation setting. Lower temperatures yield more valid molecules but with less diversity, making them preferable when generating valid molecular structures is the priority.

**Inference Speed.** An important advantage of our approach is the significantly higher sample efficiency. Language models inherently support much faster sampling rates compared to diffusion models, and advanced techniques like KV-cache (Radford et al., 2019) further accelerate this process. In our experiments, we observed that sampling 10,000 molecular structures achieved an average speed of 39.8 samples per second on a single A100 GPU with a batch size of 16. In contrast, diffusion-based models such as EDM (Hoogeboom et al., 2022) and GeoLDM (Xu et al., 2023) reached only 1.4 samples per second, representing a nearly $28\times$ speedup.

## 6 CONCLUSION AND DISCUSSION

In this work, we introduced Mol-StrucTok, a novel method for tokenizing 3D molecular structures using spherical line notation and a vector quantizer. Our approach not only demonstrated competitive chemical stability but also exhibited significantly stronger conditioning capabilities and much faster generation speeds compared to existing methods.

The first future direction is to improve the quality of the quantizers to enhance both reconstruction accuracy and downstream modeling performance. Second, as a general framework for linearizing 3D molecules and enabling language-based modeling, Mol-StrucTok opens up the potential for exploring hybrid representations. One promising direction is to integrate molecular generation with natural language, supporting joint modeling of 3D molecular structures and text, paving the way for truly native multimodal models. Another exciting potential lies in expanding the scope of 3D molecular design, where language models can facilitate more advanced conditional generation tasks, such as pocket-guided ligand design, 3D molecule editing, etc. Such advancements would enable the generation of molecules tailored to specific biological or chemical environments.

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

## A  CALCULATION OF THE LOCAL SPHERICAL COORDINATES

Consider the current atom $v_i$ with the coordinates $\boldsymbol{x}_i$ and its corresponding focal atom and reference atoms with the coordinates $\boldsymbol{x}_f, \boldsymbol{x}_{c_1}, \boldsymbol{x}_{c_2}$. We assume that $\boldsymbol{x}_f$ is the origin of the local frame. The distance $d_i$ is calculated as follows:

$$d_i = \|\boldsymbol{x}_{if}\| = \|\boldsymbol{x}_i - \boldsymbol{x}_f\|. \tag{16}$$

The vectors $\boldsymbol{x}_{c_1 f} = \boldsymbol{x}_{c_1} - \boldsymbol{x}_f$ and $\boldsymbol{x}_{c_2 f} = \boldsymbol{x}_{c_2} - \boldsymbol{x}_f$ define a plane. Since these vectors are not necessarily orthogonal, we will use the Gram-Schmidt process to orthogonalize the basis.

Let:

$$\mathbf{e_1} = \frac{\boldsymbol{x}_{c_1 f}}{\|\boldsymbol{x}_{c_1 f}\|}, \tag{17}$$

$$\mathbf{e_2} = \frac{\boldsymbol{x}_{c_2 f} - (\boldsymbol{x}_{c_2 f} \cdot \mathbf{e_1})\mathbf{e_1}}{\|\boldsymbol{x}_{c_2 f} - (\boldsymbol{x}_{c_2 f} \cdot \mathbf{e_1})\mathbf{e_1}\|}. \tag{18}$$

This gives us two orthogonal unit vectors $\mathbf{e_1}$ and $\mathbf{e_2}$ in the plane spanned by $\boldsymbol{x}_{c_1 f}$ and $\boldsymbol{x}_{c_2 f}$. The angle $\theta_i$ is the angle between the vector $\boldsymbol{x}_{if} = \boldsymbol{x}_i - \boldsymbol{x}_f$ and the normal to the plane. The normal to the plane can be obtained as:

$$\mathbf{n} = \mathbf{e_1} \times \mathbf{e_2}. \tag{19}$$

Then $\theta_i$ is computed as:

$$\theta_i = \arccos\left(\frac{\boldsymbol{x}_{if} \cdot \mathbf{n}}{\|\boldsymbol{x}_{if}\|}\right). \tag{20}$$

The angle $\varphi_i$ is the azimuthal angle in the plane spanned by $\boldsymbol{x}_{c_1 f}$ and $\boldsymbol{x}_{c_2 f}$. First, project $\boldsymbol{x}_{if}$ onto the plane:

$$\boldsymbol{x}_{if}^{\text{proj}} = \boldsymbol{x}_{if} - (\boldsymbol{x}_{if} \cdot \mathbf{n})\mathbf{n}. \tag{21}$$

Then compute $\varphi_i$ as the angle between $\boldsymbol{x}_{if}^{\text{proj}}$ and $\mathbf{e_1}$:

$$\varphi_i = \arccos\left(\frac{\boldsymbol{x}_{if}^{\text{proj}} \cdot \mathbf{e_1}}{\|\boldsymbol{x}_{if}^{\text{proj}}\|}\right). \tag{22}$$

## B  PROOF OF THE SE(3)-INVARIANCE

### B.1  TRANSLATION INVARIANCE

Translation invariance means that if we shift all points in space by the same vector, the values of $d_i$, $\theta_i$ and $\varphi_i$ should not change.

$\boldsymbol{x}, \mathbf{x}$

Let's define a translation by a vector $\boldsymbol{t}$. Under translation, all position vectors $\boldsymbol{x}_i$, $\boldsymbol{x}_f$, $\boldsymbol{x}_{c_1}$, and $\boldsymbol{x}_{c_2}$ are shifted as:

$$\boldsymbol{x}'_i = \boldsymbol{x}_i + \boldsymbol{t}, \quad \boldsymbol{x}'_f = \boldsymbol{x}_f + \boldsymbol{t}, \quad \boldsymbol{x}'_{c_1} = \boldsymbol{x}_{c_1} + \boldsymbol{t}, \quad \boldsymbol{x}'_{c_2} = \boldsymbol{x}_{c_2} + \boldsymbol{t}. \tag{23}$$

**Invariance of $d_i$**  Under translation, The distance $d_i$ becomes:

$$d'_i = \|\boldsymbol{x}'_i - \boldsymbol{x}'_f\| = \|(\boldsymbol{x}_i + \boldsymbol{t}) - (\boldsymbol{x}_f + \boldsymbol{t})\| = \|\boldsymbol{x}_i - \boldsymbol{x}_f\| = d_i. \tag{24}$$

So, $d_i$ is invariant under translation.

**Invariance of $\theta_i$**  Since translation shifts all vectors by $\mathbf{t}$, we have:

$$\boldsymbol{x}'_{c_1 f} = \boldsymbol{x}'_{c_1} - \boldsymbol{x}'_f = \boldsymbol{x}_{c_1} - \boldsymbol{x}_f, \quad \boldsymbol{x}'_{c_2 f} = \boldsymbol{x}'_{c_2} - \boldsymbol{x}'_f = \boldsymbol{x}_{c_2} - \boldsymbol{x}_f. \tag{25}$$

Therefore, the plane defined by $\boldsymbol{x}_{c_1}, \boldsymbol{x}_{c_2}$ remains the same, and the normal vector $\mathbf{n}$ (which is computed using the cross product) is unchanged under translation:

$$\mathbf{n}' = \mathbf{n}. \tag{26}$$

The vector $\boldsymbol{x}_{if}$ is also unchanged by translation:

$$\boldsymbol{x}'_{if} = \boldsymbol{x}'_i - \boldsymbol{x}'_f = \boldsymbol{x}_i - \boldsymbol{x}_f = \boldsymbol{x}_{if}.$$

Therefore, the angle $\theta'_i$, which is the angle between $\boldsymbol{x}'_{if}$ and $\mathbf{n}'$, is unchanged.

**Invariance of $\varphi_i$**  $\varphi'_i$ is the azimuthal angle between the projection of $\boldsymbol{x}'_{if}$ onto the plane and $\mathbf{e_1}'$. Since $\boldsymbol{x}'_{if}$, $\mathbf{e_1}'$, and $\mathbf{n}'$ are all unchanged under translation, the projection of $\boldsymbol{x}'_{if}$ onto the plane and the angle between $\boldsymbol{x}'^{\mathrm{proj}}_{if}$ and $\mathbf{e_1}'$ remains unchanged.

### B.2  Rotation invariance

Suppose we apply a global rotation $\mathbf{R}$ to all points. Under this transformation, each vector transforms as:

$$\boldsymbol{x}'_i = \mathbf{R}\boldsymbol{x}_i, \quad \boldsymbol{x}'_f = \mathbf{R}\boldsymbol{x}_f, \quad \boldsymbol{x}'_{c_1} = \mathbf{R}\boldsymbol{x}_{c_1}, \quad \boldsymbol{x}'_{c_2} = \mathbf{R}\boldsymbol{x}_{c_2}. \tag{27}$$

**Invariance of $d_i$**  Under rotation, we have:

$$d'_i = \|\boldsymbol{x}'_i - \boldsymbol{x}'_f\| = \|\mathbf{R}(\boldsymbol{x}_i - \boldsymbol{x}_f)\|. \tag{28}$$

Since the magnitude of a vector is invariant under rotation (rotation matrices preserve lengths):

$$d'_i = \|\boldsymbol{x}_i - \boldsymbol{x}_f\| = d_i. \tag{29}$$

Therefore, $d_i$ is invariant under rotation.

**Invariance of $\theta_i$**  Under rotation, $\boldsymbol{x}_{if}$ transform as:

$$\boldsymbol{x}'_{if} = \boldsymbol{x}'_i - \boldsymbol{x}'_f = \mathbf{R}\boldsymbol{x}_i - \mathbf{R}\boldsymbol{x}_f = \mathbf{R}\boldsymbol{x}_{if}. \tag{30}$$

The normal vector transform as:

$$\mathbf{e_1}' = \frac{\boldsymbol{x}'_{c_1 f}}{\|\boldsymbol{x}'_{c_1 f}\|} = \frac{\mathbf{R}\boldsymbol{x}_{c_1 f}}{\|\boldsymbol{x}_{c_1 f}\|} = \mathbf{R}\mathbf{e_1}. \tag{31}$$

$$\mathbf{e_2}' = \frac{\boldsymbol{x}'_{c_2 f} - (\boldsymbol{x}'_{c_2 f} \cdot \mathbf{e_1}')\mathbf{e_1}'}{\|\boldsymbol{x}'_{c_2 f} - (\boldsymbol{x}'_{c_2 f} \cdot \mathbf{e_1}')\mathbf{e_1}'\|} = \frac{\mathbf{R}\boldsymbol{x}_{c_2 f} - (\mathbf{R}\boldsymbol{x}_{c_2 f} \cdot \mathbf{R}\mathbf{e_1})\mathbf{R}\mathbf{e_1}}{\|\mathbf{R}\boldsymbol{x}_{c_2 f} - (\mathbf{R}\boldsymbol{x}_{c_2 f} \cdot \mathbf{R}\mathbf{e_1})\mathbf{R}\mathbf{e_1}\|}. \tag{32}$$

Given $\mathbf{R}$ is a unitary matrix, which $RR^T = I$. Then,

$$\boldsymbol{x}'_{c_2 f} \cdot \mathbf{e_1}' = \boldsymbol{x}'^T_{c_2 f}\mathbf{e_1}' = \boldsymbol{x}'_{c_2 f}\mathbf{R}^T\mathbf{R}\mathbf{e_1}' = \boldsymbol{x}'_{c_2 f}\mathbf{e_1}' = \boldsymbol{x}_{c_2 f} \cdot \mathbf{e_1}. \tag{33}$$

Thus, we can obtain that:

$$\mathbf{e_2}' = \mathbf{R}\mathbf{e_2}. \tag{34}$$

$$\mathbf{n}' = \mathbf{e_1}' \times \mathbf{e_2}' = \mathbf{R}\mathbf{n}. \tag{35}$$

$$\cos\theta'_i = \frac{\boldsymbol{x}'_{if} \cdot \mathbf{n}'}{\|\boldsymbol{x}'_{if}\|} = \frac{\mathbf{R}\boldsymbol{x}_{if} \cdot \mathbf{R}\mathbf{n}}{\|\boldsymbol{x}_{if}\|} = \frac{\boldsymbol{x}_{if} \cdot \mathbf{n}}{\|\boldsymbol{x}_{if}\|}. \tag{36}$$

Hence, $\theta'_i = \theta_i$ is invariant under rotation.

**Invariance of $\varphi_i$**  Under rotation, both the projection $\boldsymbol{x}'^{\mathrm{proj}}_{if}$ transform as:

$$\boldsymbol{x}'^{\mathrm{proj}}_{if} = \boldsymbol{x}'_{if} - (\boldsymbol{x}'_{if} \cdot \mathbf{n}')\mathbf{n}' = \mathbf{R}\boldsymbol{x}_{if} - (\mathbf{R}\boldsymbol{x}_{if} \cdot \mathbf{R}\mathbf{n})\mathbf{R}\mathbf{n}' = \mathbf{R}\boldsymbol{x}^{\mathrm{proj}}_{if}. \tag{37}$$

Thus,

$$\cos\varphi'_i = \frac{\boldsymbol{x}'^{\mathrm{proj}}_{if} \cdot \mathbf{e_1}'}{\|\boldsymbol{x}'^{\mathrm{proj}}_{if}\|} = \frac{\mathbf{R}\boldsymbol{x}^{\mathrm{proj}}_{if} \cdot \mathbf{R}\mathbf{e_1}}{\|\boldsymbol{x}^{\mathrm{proj}}_{if}\|} = \frac{\boldsymbol{x}^{\mathrm{proj}}_{if} \cdot \mathbf{e_1}}{\|\boldsymbol{x}^{\mathrm{proj}}_{if}\|}. \tag{38}$$

Therefore, $\varphi'_i = \varphi_i$ is invariant under rotation.

## C IMPLEMENTATION DETAILS

### C.1 VECTOR QUANTIZATION

After obtaining the descriptors, we applied normalization to all of them. For the length descriptors, we used log normalization, while for the angle descriptors, we normalized them to a $0 - 1$ range within the $0 - \pi$ space. This normalization proved beneficial for training the quantizer. During training, we used a batch size $512$, learning rate of $1e - 4$ with a 5-epoch warm-up, and the learning rate schedule remained constant throughout.

### C.2 GENERATION

For training, we set the batch size to $64$ and trained for $200$ epochs with a learning rate of $4e - 4$. We applied a warm-up phase of $3000$ steps, followed by a linear decay schedule. The generation configuration used a default temperature of $0.7$, top-k sampling with $k = 50$, and a maximum sequence length of $100$, while the longest sequence in the dataset was $77$ tokens. We applied a repetition penalty of $1$ to avoid redundant generations.

During the reconstruction of coordinates, since the molecular topology is generated simultaneously, we employed a topology-aware optimization algorithm to refine the 2D structure. This optimization follows the approach used in LDM-3DG (You et al., 2023).

### C.3 UNDERSTANDING

In Graphormer, there are two input embeddings: an atom type embedding and a degree embedding. In our implementation, we additionally incorporate Mol-StrucTok's embedding while keeping the other components unchanged. During training, we set the learning rate to $1.5e - 4$ with a linear decay schedule, using a batch size of $16$ and a weight decay of $0.01$.

## D FURTHER ANALYSIS ON STRUCTURAL ALPHABETS.

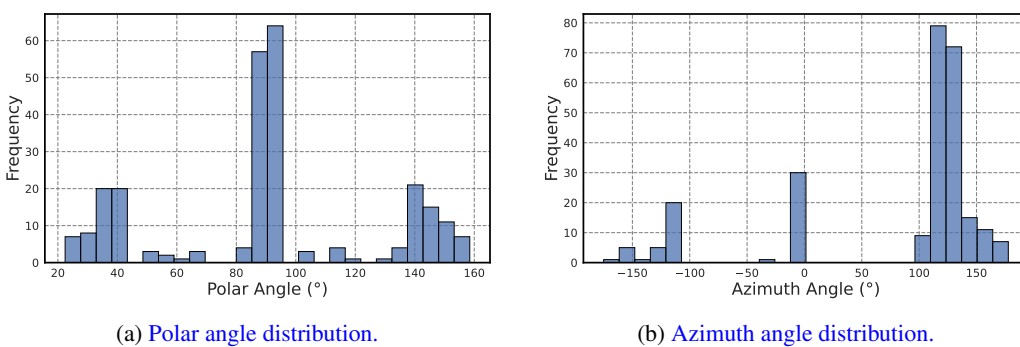

(a) Polar angle distribution.

(b) Azimuth angle distribution.

Figure 5: Decoding generation descriptor distributions.

**Decoding on angular descriptors.** In addition to the distance, the spherical coordinate system also includes the polar and azimuthal angles. We computed the same distribution as in Figure 4a for the polar and azimuthal angles, as shown in Figure 6 5. We decoded the distribution of the 256 tokens with respect to these two features. Although the exact meanings of the tokens in the vocabulary are not readily interpretable, it is evident that both the polar and azimuthal angles exhibit clear peaks. This suggests that the angle representations derived from our local spherical coordinates are concentrated. Furthermore, VQVAE is able to adaptively learn this concentration, allocating more tokens to regions with higher density.

**Atom type and shared alphabets.** A key feature of our VQVAE design is the decoupling of atom types and local structural environments. Some vocabulary indices are capable of representing the local environments of all atom types, while others are specific to a single atom type. This design

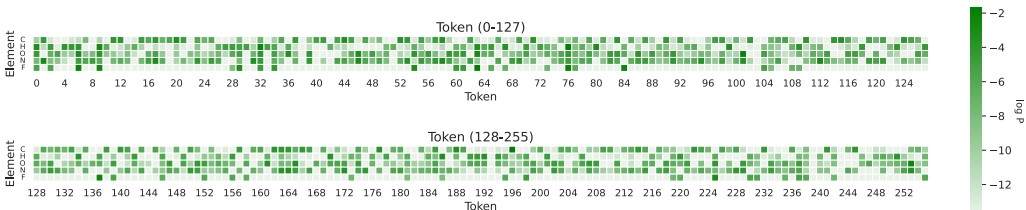

Figure 6: Heatmap of atom type hit ratios across shared alphabets. The darker the color, the higher the occurrence probability (log P).

| RMSD | Gen-Length | Gen-Polar | Gen-Azimuth |
|---|---|---|---|
| Mol-StrucTok | 0.01 | 0.06 | 0.06 |
| - w.o. feature normalization | 0.03 | 0.12 | 0.11 |
| - w.o. sign prediction | 0.01 | 0.08 | 0.84 |

Table 7: Performance comparison of Mol-StrucTok and its ablations.

enables different atom types to share the same local structural descriptors, resulting in a more compact representation. To evaluate this property, we analyzed the distribution of different atom types across each token ID in the PCQM4Mv2 dataset. The results are visualized in the heatmap (Figure 6). The heatmap reveals that the learned vocabulary is highly utilized, with nearly all token IDs being assigned to meaningful local structures. Furthermore, the majority of local structures are shared among different atom types, demonstrating the generalization capability of our approach. Only a small number of token IDs, such as 19, 30, and 33, are specific to the local environments of particular atom types. This suggests that our design effectively balances compactness and specificity in structural representation.

**Ablation study.** In our implementation, the model architecture adopts an MLP structure similar to FoldSeek (van Kempen et al., 2022) but incorporates fine-tuned hyperparameters and an additional head dedicated to predicting the sign of the azimuthal angle. This enhancement plays a critical role in improving the reconstruction of azimuthal angles. Furthermore, we normalize length features to the log space and angle features to the range of $[0, 1]$. Such normalization ensures that the model learns descriptors with consistent scales, thereby facilitating the VQ-VAE training process. To further validate our approach, we conducted additional ablation studies on the generation descriptors. Specifically, we evaluated reconstruction errors using RMSD for length, polar angle, and azimuthal angle predictions in 7.

