# OpenReview forum: "Tokenizing 3D Molecule Structure with Quantized Spherical Coordinates"
_ICLR.cc/2025/Conference — Submitted to ICLR 2025_

### Official Review · Reviewer_8GHQ · 2024-11-03

**Soundness:** 2
**Presentation:** 2
**Contribution:** 2
**Rating:** 5
**Confidence:** 4

**Summary:**

This work proposes a novel language model based 3D molecule generation framework. Novel spherical coordinate based 3D line notation method combined with VQ-VAE model is adopted to tokenize 3D molecules, and GPT-2 models are applied to this tokenization framework. Benchmark experiments are conducted to show the promising performance of the proposed method.

**Strengths:**

Originality:
This work makes a good originality contribution of  proposing a novel 3D molecule tokenization method that is applicable for language models.

Quality:
This work presents sufficient theoretic analysis and proofs for the proposed 3D tokenization method, and benchmark experiments are conducted to demonstrate the promising performance.

Clarity:
The writing of this paper is fairly good. The paper gives sufficient description to help readers understand the key pipeline and steps of 3D molecule generation.

Significance:
The proposed method is useful in leveraging the power of large language models to the generation and modeling of complicated 3D molecule structures.

**Weaknesses:**

(1) Some important methods details are not clearly clarified or presented. Authors are encouraged to add explanations to the following questions:
- How the order of atom tokens in equation (2) is decided? By BFS or DFS traverse algorithm on 2D topology graphs?
- The $x_i$ in line 266 is actually $g_i$ in equation (7)?
- Why the sign of torsion angle $sign(\varphi_i)$ appears in the generation descriptor of equation (7) but not in the combined descriptor of equation (9)?
- Why local atomic environment (line 266-267) is needed? Can authors discuss the negative impacts of not including it and conduct ablation studies if possible?

(2) In benchmark experiments, an important baseline Geo2Seq [1] (it also proposes a spherical coordinate based 3D molecule tokenization framework) is not presented and compared. Particularly, Geo2Seq has higher molecule stability than authors' method, so more discussion about differences and advantages over Geo2Seq is needed.

(3) VQ-VAE based vector quantization is applied to atom descriptor in equation (9) to form the final generation target. A natural question is what is the advantages of VQ-VAE over simple discretization (i.e., map floating numbers to pre-split bins, e.g., 0-30, 30-60, 60-90, ... for angles) ? Authors are encouraged to discuss different possible quantization methods and conduct ablation experiments to justify the advantages of VQ-VAE.

[1] Li, Xiner, et al. "Geometry Informed Tokenization of Molecules for Language Model Generation." arXiv preprint arXiv:2408.10120 (2024).

**Questions:**

See Weaknesses part.

**Details Of Ethics Concerns:**

No ethics concerns.

---

> ### Author Response · Authors · 2024-11-22
> **Response to Reviewer 8GHQ (1/2)**
>
> We sincerely thank the reviewer for their constructive feedback and valuable suggestions.
>
> > W1.1: Atom token ordering
> >
>
> We did not design a specific algorithm for determining the order of atom tokens; instead, we used the default canonicalization in RDKit, which employs a DFS-like algorithm. Additionally, it is important to note that our method is not dependent on any particular order—random orders can also be applied.
>
> > W1.2, 1.3: Paper typo
> >
>
> Thanks for pointing out our typos. We have fixed in our revised manuscript.
>
> > W1.4:  Lack of justification of local atomic environment
> >
>
> Our goal is to develop a **general-purpose molecular tokenizer**, not solely focused on molecule generation. Understanding the 3D spatial neighborhood structure is crucial for capturing the local atomic environment, which plays a significant role in understanding tasks. However, the generation features themselves do not consider information about neighboring atoms. Therefore, we include the understanding features to incorporate this information. Removing these features simplifies the task and could improve performance on generation tasks. However, this comes at the cost of diminished capability for enhancing understanding tasks, undermining the tokenizer's broader applicability.
>
> > W2: Lack of comparison with Geo2Seq
> >
>
> Firstly, from a procedural standpoint (as per [ICLR Reviewer Guidelines](https://iclr.cc/Conferences/2025/ReviewerGuide)), Geo2Seq was uploaded to arXiv on August 19, and it has not been published in a peer-reviewed venue. Therefore, it is not mandatory to treat Geo2Seq as a baseline. Nevertheless, we have appropriately cited and discussed their work in the paper. Additionally, Geo2Seq has not provided open-source code. Based on the results reported in their paper, our method achieves comparable performance when utilizing GPTs as LM.
>
> | Unconditional | Atom Sta | Mol Sta | Valid | Valid & Unique |
> | --- | --- | --- | --- | --- |
> | Geo2Seq | 98.3% | **90.3%** | 94.8% | 80.6% |
> | Mol-StrucTok | **98.5%** | 88.3% | **98.0%** | **85.1%** |
>
> | Conditional | alpha | gap | homo | lumo | mu | Cv |
> | --- | --- | --- | --- | --- | --- | --- |
> | Geo2Seq | 0.46 | 98 | **57** | 71 | **0.164** | 0.275 |
> | Mol-StrucTok | **0.33** | **89** | 64 | **62** | 0.285 | **0.169** |
>
> We can, however, further discuss the limitations of Geo2Seq:
>
> 1. **Lack of Generalization in Global Spherical Coordinates**: Geo2Seq's use of global spherical coordinates makes it difficult to generalize across scales.
> 2. **Inefficiency of Number-Based Tokenization**: Directly tokenizing numerical values results in several issues:
>     1. Precision is bounded by the number of decimal places, and inconsistent decimal precision hinders generalization.
>     2. Segmenting numbers into tokens often produces long sequences, especially for large numbers or high-precision decimals. This increases computational burden and complicates modeling.
>
> In contrast, our approach employs discretization (e.g., VQVAE), which significantly reduces the length of numerical representations and computational overhead.
>
> Moreover, our method is designed as a **general-purpose tokenizer.** We pretrain the vocabulary on the PCQM dataset, ensuring generalization across diverse datasets. Geo2Seq, however, relies on hash-based discretization trained only on specific datasets, limiting its generalizability. Furthermore, Geo2Seq does not serialize bond features, which are crucial for representing 2D graphs in real-world scenarios (Geo2Seq relies on RDKit to reconstruct chemical bonds). Finally, our method includes **understanding features**, which Geo2Seq lacks. While these design choices for generalization may slightly reduce performance on generation tasks, they expand the applicability of our method to a broader range of scenarios.

---

> ### Author Response · Authors · 2024-11-22
> **Response to Reviewer 8GHQ (2/2)**
>
> > W3: Pre-split bin-based tokenization
> >
>
> Thanks for your suggestion. While pre-split bins can discretize continuous values, they cannot reconstruct the original continuous descriptors from the discretized bins. This makes it impossible to restore molecular structures. In contrast, VQ-VAE allows for the reconstruction of continuous descriptors from discrete tokens, preserving the structural information.
>
> If we were to use the midpoint of each bin as the reconstructed continuous value, we can estimate the vocabulary size required to achieve reconstruction errors comparable to those of VQ-VAE. For simplicity, let us consider only the generation descriptors: In our method, VQ-VAE is trained jointly on understanding and generation descriptors. The reconstruction RMSD for generation descriptors is approximately 0.01Å  (with a range of 0.9Å∼2.7Å), with angular errors of 0.06 (with ranges of [0,π] and [−π,π] for elevation and azimuthal angles, respectively). These can be considered upper bounds for the reconstruction errors achievable by VQ-VAE when trained only on generation descriptors.
>
> To match this error using pre-split bins under the assumption of uniform data distribution: The bin interval for distances must be 0.04 Å, corresponding to 45 bins. The bin interval for angles must be 0.24, corresponding to 13 bins for elevation and 26 bins for azimuth. Taking the Cartesian product of these bins yields a vocabulary size of 45×13×26=15,210, which far exceeds the VQ-VAE vocabulary size of 256.
>
> This illustrates that VQ-VAE adaptively learns an optimal vocabulary, achieving better compression performance while maintaining low reconstruction error. Such adaptability is crucial for effectively handling diverse molecular datasets and achieving compact yet accurate representations.
>
> We perform additional experiments on pre-split-based discretization as described above, better with using VQ-VAE.
>
> | Unconditional | Atom Sta | Mol Sta | Valid | Valid & Unique |
> | --- | --- | --- | --- | --- |
> | Pre-split bins | 89.9% | 35.6% | 96.1% | **91.4%** |
> | Mol-StrucTok | **98.5%** | **88.3%** | **98.0%** | 85.1% |

---

> > ### Comment · Reviewer_8GHQ · 2024-11-24
> > **Follow-up Response**
> >
> > I appreciate authors' efforts in rebuttal. Some of my concerns have been addressed so I raised my rating. Nonetheless, I am not completely convinced by the advantages of the proposed method over Geo2Seq claimed by authors. While VQVAE may be better than number-based tokenization intuitively, advantages of global spherical coordinates are not fully demonstrated by experiments, particularly given that Mol-StrucTok underperforms Geo2Seq in Molecule stability. I think more experiments, e.g., large molecule benchmark experiments, are needed to give stronger signals. Hence, in the current state, I still vote for rejection.

---

> > > ### Author Response · Authors · 2024-11-25
> > > **Follow-up Response to Reviewer 8GHQ**
> > >
> > > We sincerely appreciate your acknowledgment of the improvements in our rebuttal and your thoughtful feedback. We address your remaining concerns regarding the comparison with Geo2Seq below.
> > >
> > > 1. **Non-mandatory comparison with Geo2Seq.**
> > >
> > > To begin with, we respectfully maintain that, as noted in the [ICLR Reviewer Guide](https://iclr.cc/Conferences/2025/ReviewerGuide), a direct comparison with concurrent work is not obligatory, particularly when Geo2Seq is unpublished and its code is unavailable. This lack of access to implementation details makes a fair, systematic comparison infeasible. However, we have strived to provide a detailed conceptual discussion in our rebuttal (see our response to W2) and further experiments on related setups as feasible within the review timeline.
> > >
> > > 2. **Global vs. Local Spherical Coordinates.**
> > >
> > > We understand your concern regarding the advantages of global spherical coordinates over local spherical coordinates. To address this, we implemented a version of our model using global spherical coordinates, as mentioned in Geo2Seq. All other design elements were kept consistent with the local spherical coordinate implementation. Our experiments reveal that global coordinates underperform on node-level VQVAE under reconstruction task, since the context of each atom is not provided. This subsequently impacts downstream generation tasks, as shown in the following results:
> > >
> > > | reconstruction RMSD | und-length | und-angle | gen-length | gen-polar | gen-azimuth |
> > > | --- | --- | --- | --- | --- | --- |
> > > | global sph coord | 0.03 | 0.09 | 0.07 | 0.18 | 0.19 |
> > > | local sph coord | 0.02 | 0.07 | 0.01 | 0.06 | 0.06 |
> > >
> > > | Unconditional | Atom Sta | Mol Sta |
> > > | --- | --- | --- |
> > > | global | 64.2% | 37.6% |
> > > | local | 98.5% | 88.3% |
> > >
> > > These results demonstrate that local spherical coordinates provide significant advantages in both reconstruction accuracy and generation quality, indicating that local spherical coordinates are better suited for our design.
> > >
> > > 3. **Additional Experiments.**
> > >
> > > We have initiated experiments on the GEOM-Drug dataset to explore the performance on larger molecules as suggested. However, due to the time constraints and the preprocessing required, these results are still in progress. We commit to including these in an extended version of the paper.
> > >
> > > 4. **Practical Considerations.**
> > >
> > > Our unconditional generation results achieve 88.3% molecule stability, which we believe is sufficient for practical scenarios where sampling can produce stable molecules. Additionally, our approach captures 2D topology information, mitigating concerns about minor 3D structural errors that could affect bond connectivity. More importantly, our method offers strong controllable generation capabilities, where we match or surpass Geo2Seq in key metrics, also demonstrating significant performance.
> > >
> > > In conclusion, while Geo2Seq is a relevant concurrent work, we believe our design choices and results provide compelling evidence of the strengths of our method as a general-purpose molecular tokenizer. We hope these clarifications address your concerns.

---

> > > > ### Comment · Reviewer_8GHQ · 2024-11-26
> > > > **Follow-up Response**
> > > >
> > > > I appreciate authors' latest response. However, global vs. local spherical coordinates comparison results look strange to me. To my understanding, the results show that VQVAE + global spherical coordinates performs significantly worse than Geo2Seq (number-based tokenization + global spherical coordinates) and Mol-StrucTok (VQVAE + local spherical spherical coordinates). Why combining VQVAE with global coordinates causes such a significant performance drop while using global spherical coordinates does not impact the performance in Geo2Seq (Note Geo2Seq reports reasonable performance results on QM9 and GEOM-DRUGS benchmarks)? I hope authors could provide some explanations.

---

> > > > > ### Author Response · Authors · 2024-11-27
> > > > > **Follow-up Response to Reviewer 8GHQ**
> > > > >
> > > > > Dear Reviewer 8GHQ,
> > > > >
> > > > > Thank you for your further feedback. We would like to clarify some misunderstandings about our method and its comparison with Geo2Seq. These distinctions are essential for understanding why the observed performance between ours and Geo2Seq cannot be directly or jointly compared. As noted in Section W2, we expand on these differences below for further clarity:
> > > > >
> > > > > 1. **Different Structural Modeling Approaches**: Geo2Seq employs **global spherical coordinates**, whereas our method is based on **local spherical coordinates**.
> > > > > 2. **Different Discretization Techniques**: Geo2Seq uses **number-based tokenization**, while we employ a **atom-level VQVAE-based approach**.
> > > > > 3. **Additional Understanding Descriptors**: Our framework incorporates **understanding descriptors**, which are crucial for molecular understanding tasks but may slightly compromise performance in generation tasks.
> > > > > 4. **Distinctive Task Complexity**: Our primary contribution lies in introducing a 3D **line notation** that integrates 2D graph representations of molecules. This joint representation increases task complexity in GPT modeling, as our task involves **jointly generating 2D and 3D molecular structures**, whereas Geo2Seq focuses solely on 3D generation.
> > > > >
> > > > > > **Explanation for Performance Drop**
> > > > > >
> > > > >
> > > > > Regarding the performance drop when combining VQVAE with global spherical coordinates, we offer the following explanation:
> > > > >
> > > > > As highlighted in the reconstruction error analysis from our last rebuttal, using **global spherical coordinates** in an atom-level VAE setting significantly increases learning difficulty. This is evident from the **3-7x higher reconstruction RMSD** in the generation descriptors. Since our VQVAE inputs are atom-based, capturing **local descriptors** is inherently simpler. In contrast, independently learning atomic global coordinates without explicit access to the molecule's overall structure poses a significant challenge, which subsequently affects downstream task performance.
> > > > >
> > > > > > **Final Remarks**
> > > > > >
> > > > >
> > > > > We also wish to emphasize that while Geo2Seq is a commendable contribution, we kindly request the reviewer to consider that comparative results with concurrent works should not necessarily lead to rejection, especially given the **novelty and unique contributions** of our approach. Our work introduces a new paradigm for 3D molecular modeling, which we believe provides valuable insights and advancements to the field.
> > > > >
> > > > > ---
> > > > > Thank you for taking the time to review our work. We hope this response sufficiently addresses your concerns and highlights the value and contributions of our research. Since the discussion period is approaching the end, we would greatly appreciate your reconsideration of the score for our submission.
> > > > >
> > > > > Sincerely,
> > > > >
> > > > > Authors

---

> ### Author Response · Authors · 2024-12-02
> **kindly ask for response towards our explanation**
>
> Dear reviewer 8GHQ,
>
> Towards your last message, we have provided the following response several days ago, please refer here again: https://openreview.net/forum?id=UqrSyATn7F&noteId=SsmAmGbExG.
>
> However, unluckily, though we provided explanations and responses to your questions, we haven't received your further comments and feedback about our message.
>
> **We do try our best to give you thoughtful explanations and we do look forward your engagement to review our response.**
>
> Sincerely look forward to your comment.
>
> Best,
>
> Authors.

---

### Official Review · Reviewer_8Laj · 2024-11-04

**Soundness:** 3
**Presentation:** 4
**Contribution:** 3
**Rating:** 8
**Confidence:** 3

**Summary:**

The authors express 3D structure using spherical coordinates, then tokenize them with a VQ-VAE. The tokens are then used for training a GPT-2 model for 3D molecular generation tasks. They also use the discrete representations in a Graphormer model for property prediction on QM9, and observe consistent improvements across various molecular properties.

**Strengths:**

1. The authors conduct an extensive set of experiments. They measure validity+uniqueness of generated molecules with different bond assignment methods, perform PoseBusters tests, evaluate quantum mechanical properties, and measure MAE for QM9 property prediction. They achieve state-of-the-art results in most experiments.
2. They also perform additional analysis regarding the inference speed of their method and the effect of the generation temperature on balancing quality and diversity.

**Weaknesses:**

1. This is a hand-crafted tokenization scheme and should be compared to other tokenizers (e.g. BPE-based tokenizers), not just diffusion models and MPNN-based methods.
2. It may also be helpful to compare with structures expressed in other coordinate systems. I'd imagine that without SE(3) invariance there would be a wider range of possible tokenized sequences, making it harder for the GPT-2 model to learn.

**Questions:**

1. To the best of my knowledge, PoseBusters has an energy ratio test, which tends to be very difficult to satisfy. Have you tried on that test? I think that test would be more meaningful since most methods already perform very well on the tests you demonstrated in Table 2.

2. What's the motivation for a purely MLP-based quantized auto-encoder architecture? Most other autoencoder-based tokenizers use some form of GNN or sequence model. What are the tradeoffs between element-wise discretization and discretization with message propagation?

---

> ### Author Response · Authors · 2024-11-22
> **Response to Reviewer 8Laj**
>
> We sincerely thank the reviewer for their positive and encouraging feedback.
>
> > W1: Alternative BPE-based tokenization methods.
> >
>
> Thank you for pointing this out. Tokenization is indeed an interesting perspective. Our paper primarily focuses on the serialization and discretization of 3D molecules, which is why we defaulted to using the SELFIES token-based tokenizer. During the rebuttal, we also implemented a BPE-based tokenizer. The results, as shown in the table below, indicate that the choice of tokenization method has only a minor impact on performance.
>
> |  | Atom Sta | Mol Sta | Valid |
> | --- | --- | --- | --- |
> | bpe | 98.4% | **88.9%** | **98.6%** |
> | token-based | **98.5%** | 88.3% | 98.0% |
>
> > W2: Alternative coordinate systems.
> >
>
> You are correct that using other coordinate systems can make the task more challenging to learn. To demonstrate this, we have conducted experiments using Euclidean coordinates. The results show that the LM struggles to generate stable molecules under this representation. In contrast, the spherical  coordinate system constrains the numerical range, making it easier for the model to learn effectively.
>
> |  | Atom Sta | Mol Sta | Valid |
> | --- | --- | --- | --- |
> | Euclidean | 13.0% | 0.0% | **99.3%** |
> | Spherical | **98.5%** | **88.3%** | 98.0% |
>
> > Q1: PoseBusters' energy ratio test
> >
>
> The **Reasonable Internal Energy** term is actually the metric you mentioned. We directly used the test script from **Symphony**. We're sorry that we didn't elaborate on it in the paper.  We have updated the detailed description of the metric in our paper.
>
> > Q2: Unclear motivation of MLP-based autoencoder
> >
>
> The motivation for adopting a purely MLP-based quantized auto-encoder architecture is twofold:
>
> 1. The task itself is not particularly complex, and node-level modeling already achieves excellent RMSD performance.
> 2. We aim for the discretized structure vocabulary to exhibit generalizability across different atomic types in different molecules rather than being tailored to individual molecules. To further illustrate this, we have included additional heatmap visualizations in the Appendix.

---

### Official Review · Reviewer_7HbA · 2024-11-04

**Soundness:** 3
**Presentation:** 2
**Contribution:** 3
**Rating:** 5
**Confidence:** 4

**Summary:**

The authors introduce a novel method for tokenizing 3D molecular structures. The authors design a 2D line notation for 3D molecules by extracting the local atomic spherical system, then employ VQ-VAE to tokenize upon this 2D line notation to tokenize the spherical coordinates. Results show the competitive performance compared to several SOTA methods.

**Strengths:**

The combination of spherical line notation with vector quantization enables language models to process complex 3D data, which is challenging to discretize. This approach stands out from traditional graph-based or continuous-coordinate models by providing a discrete representation for language models without losing SE(3)-invariant information. Particularly, the augmented tokens incorporate both generation and understanding descriptors, including local spherical coordinates, bond lengths, and angles, allowing a better capture of molecular topology and spatial arrangements.

**Weaknesses:**

### Major
The authors should clarify the rationale behind selecting exactly four neighbors for the atomic descriptor and explicitly address how the descriptor $\mathbf{z}_i$ is defined for atoms with fewer than four neighbors. This is essential, as molecules with varying coordination environments will likely have different numbers of neighbors, impacting the generality of the descriptor across datasets.

### Minor
1. The paper’s notations are somewhat inconsistent and could benefit from simplification and unification. For instance, symbols like $\mathcal M$ are used only within specific sections, such as Section 3.1, and do not appear elsewhere. Additionally, the notation in Appendix C is different from that in Section 3.1. A more streamlined notation would improve readability and coherence across sections.
2. Line 177, the phrase should be rephrased as "we tokenized the molecular graph into a sequence of atom tokens $\mathbb A$ and non-atom tokens $\mathbb B$".
3. In Line 248 and Line 268, references are made to "Section X" and "Figure X," which seem placeholders. These should be updated with specific section and figure numbers
4. Line 239, "Inspired by previous works" is ambiguous. The authors should specify which studies or methods they are referring to here.

**Questions:**

See weaknesses.

---

> ### Author Response · Authors · 2024-11-22
> **Response to Reviewer 7HbA**
>
> We sincerely thank the reviewer for their time and effort in reviewing our manuscript.
>
> > W1: Unclear rationale and handling of fixed neighbor selection
> >
>
> We chose four neighbors for the atomic descriptor in the understanding task because **using only three atoms is insufficient to capture the local environment effectively**. The most common atom, carbon, typically has four neighbors, and increasing this number beyond four yields similar performance. We wanted to limit the feature dimensionality of the understanding descriptors to ensure it does not overshadow the generation descriptors during reconstruction.
>
> While our method cannot handle molecules with fewer than four neighbors, such cases are **exceedingly rare** (approximately 0.03% in the QM9 dataset, assuming hydrogen is included). For these rare corner cases, we opted to simply remove them from our experiments.
>
> > W2: Inconsistent and unclear notations.
> >
>
> Thank you for your suggestion. We agree that consistency in notation is important. However, we believe that the use of M is essential for maintaining the clarity and flow of the discussion in Section 3.1, so we have decided to retain it. On the other hand, we agree that the notational differences in Appendix C could cause confusion. Since removing Appendix C does not impact the completeness of the paper, we have opted to remove that section to ensure overall coherence.
>
> > W3, W4: Paper typo
> >
>
> Thanks for your careful review. We have updated these notions in our revised manuscript.
>
> > W5: References to prior works
> >
>
> Thank you for your suggestion. In the revised manuscript, we have clarified this statement by providing a detailed description and appropriate citations of the previous methods that inspired our work.

---

> > ### Author Response · Authors · 2024-11-27
> > **Looking Forward to Your Valuable Feedback**
> >
> > Dear Reviewer 7HbA,
> >
> > We sincerely thank you for your time and effort in reviewing our paper. Your feedback has been invaluable in helping us refine and improve the quality of our work.
> >
> > We have carefully addressed all the concerns and comments you raised in our previous responses. As the discussion phase deadline is approaching, we would be truly grateful if you could kindly prioritize providing your feedback at your earliest convenience.  Additionally, you are welcome to review the discussions with other reviewers if it may be helpful.
> >
> > Thank you again for your thoughtful suggestions. We look forward to any additional feedback or discussions you may have.
> >
> > Sincerely,
> >
> > Authors

---

> ### Author Response · Authors · 2024-12-01
> **Deadline is approaching**
>
> Dear Reviewer 7HbA,
>
> Thank you once again for your valuable feedback. With only **two days** remaining until the discussion period ends, we kindly request confirmation that our responses have addressed all of your concerns. We are open to any further discussions if needed.
>
> We truly understand how time-consuming and challenging the process of reviewing papers can be. We sincerely appreciate the effort each reviewer puts into reading and providing suggestions, as it is through your insights that the quality of our work improves.
>
> At the same time, as an author submitting to this conference, we fully recognize the immense dedication involved in preparing a manuscript and responding to each reviewer’s queries. The paper we submitted, along with the responses we have provided, reflects the hard work and commitment of our team. It is of great significance to us, and we simply hope for a fair and constructive evaluation.
>
> Thank you for your understanding and time. We look forward to your feedback and wish you all the best.
>
> Best regards,
>
> Authors

---

### Official Review · Reviewer_gMZe · 2024-11-05

**Soundness:** 3
**Presentation:** 3
**Contribution:** 2
**Rating:** 5
**Confidence:** 5

**Summary:**

This paper focuses on the task of 3D molecule generation with language models (LMs). The main step is to convert 3D molecules to sequences of discrete tokens. The authors first use SMILES (or SELFIES) to convert a molecule's atom and bond information to a sequence. They then use spherical coordinates and VQ-VAE to convert 3D information to discrete tokens. After obtaining these 3D molecule sequences, they train a GPT-style model for molecule generation by next-token-prediction. The model is evaluated on the random generation and conditional generation tasks.

**Strengths:**

1. This paper is well-written and easy to follow, with clear and informative tables and figures.
2. The proposed method performs well, especially on the conditional generation task.
3. The ablation study is thorough and provides useful insights.

**Weaknesses:**

1. The proposed method is quite similar to existing methods, such as FoldSeek and FoldToken. Specifically, similar to the SE(3)-invariant spherical coordinates here, FoldSeek also uses distances and angles computed based on reference nodes as SE(3)-invariant representations. In addition, Furthermore, both methods employ VQ-VAE to learn discrete tokens. These overlapping components limit the novelty of this work.
2. About the datasets: the proposed method is only evaluated on QM9 dataset, which includes only small molecules. It will be better to evaluate on larger molecules, such as geom-drug dataset.
3. More LM-based baseline methods should be included and compared, such as Geo2Seq and BindGPT.
4. Further analysis and potential visualizations of the learned structural alphabet, for example, which types of local structures can be mapped to the same code? Refer to Figure S2 and S3 of FoldSeek.


Other questions:
1. line 52: SE(3) invariance is invariance under rotation and translation, but not reflection
2. What is the model size?

**Questions:**

See weaknesses

---

> ### Author Response · Authors · 2024-11-22
> **Response to Reviewer gMZe**
>
> We thank the reviewer for their time and detailed feedback on our manuscript.
>
> > W1: Lack of novelty due to similarities with existing methods
> >
>
> We think that your comment regarding novelty is somewhat biased, so we would like to reiterate our contribution: our primary innovation lies in serializing and discretizing 3D molecular structures to support language model-based modeling. We openly acknowledge that our work is inspired by FoldSeek and thoroughly discuss FoldSeek and related methodologies in the *Introduction*, *Related Work*, and *Experiments* sections of our paper. However, applying similar ideas in FoldSeek is far from trivial on molecules. To clarify the differences between our method and FoldSeek:
>
> - **Serialization of 3D Structure**: While FoldSeek operates on protein structures, which are naturally sequential, **molecules are graphs with no inherent order between atoms**. Serializing the 3D structure of molecules is not trivial. Our approach requires constructing SE(3)-invariant descriptors, which involves using both 1D line notation and 2D graph representations to achieve robust local structural encoding. This is a key distinction from FoldSeek, FoldToken, and previous AR methods for 3D molecule generation, as discussed in Section 3.2.
> - **Distinct Features**: We diverge from FoldSeek in our feature design. Our model architecture utilizes a similar MLP structure, but with fine-tuned hyperparameters and **an additional head** to predict the sign of the azimuthal angle, which significantly aids in the reconstruction of azimuthal angles We also **normalize length features** to the log space and angle features to the 0-1 range. This normalization ensures that the model learns descriptors with consistent scales, which also benefits the VQ-VAE learning process.
> - **Training Objective**: FoldSeek uses a Gaussian negative log-likelihood loss to model the distributions of aligned amino acids. In contrast, our goal is to **reconstruct the local structure of source atoms**, so we use MSE loss, which better reflects the quality of the reconstruction.
>
> These design choices collectively highlight the distinctiveness of our approach compared to FoldSeek and FoldToken, particularly in how we handle molecular structure serialization, feature processing, and training objectives.
>
> > W2: More benchmark results
> >
>
> Thank you for your suggestion. We have already started conducting additional experiments on GEOM-Drug dataset. However, due to limited computational resources, it may take some time to pre-processing and complete these experiments. We will include the results as soon as they are available.
>
> > W3: Lack of comparison with Geo2Seq and BindGPT
> >
>
> Firstly, from a procedural standpoint (as per [ICLR Reviewer Guidelines](https://iclr.cc/Conferences/2025/ReviewerGuide)), Geo2Seq was uploaded to arXiv on August 19, and neither Geo2Seq nor BindGPT has been published in a peer-reviewed venue. Therefore, it is not mandatory to treat Geo2Seq as a baseline. Nevertheless, we have appropriately cited and discussed their work in the paper. Additionally, Geo2Seq has not provided open-source code. Based on the results reported in their paper, our method achieves comparable performance when utilizing GPTs as LM.
>
> | Unconditional | Atom Sta | Mol Sta | Valid | Valid & Unique |
> | --- | --- | --- | --- | --- |
> | Geo2Seq | 98.3% | **90.3%** | 94.8% | 80.6% |
> | Mol-StrucTok | **98.5%** | 88.3% | **98.0%** | **85.1%** |
>
> | Conditional | alpha | gap | homo | lumo | mu | Cv |
> | --- | --- | --- | --- | --- | --- | --- |
> | Geo2Seq | 0.46 | 98 | **57** | 71 | **0.164** | 0.275 |
> | Mol-StrucTok | **0.33** | **89** | 64 | **62** | 0.285 | **0.169** |
>
> > W4: Further FoldSeek-like visualization
> >
>
> Our approach differs fundamentally from FoldSeek. FoldSeek is trained not for reconstruction but to learn aligned distributions, employing a Gaussian negative log-likelihood loss. In their case, a single code decodes into a distribution, which allows visualizations like Figures S2 and S3 in their work.
>
> In contrast, our task focuses on reconstruction, where the generation descriptors are designed to restore the molecular structure. Each descriptor can only be mapped to a vector, meaning visualizing atoms grouped under the same code does not provide meaningful insights for our method.
>
> > Q1: Mistaken description of SE(3)-invariance
> >
>
> We truly appreciate you pointing this out. We acknowledge that it was our mistake. The issue has been rectified in the revised manuscript.
>
> > Q2: Model size
> >
>
> The model sizes used in our method are as follows:
>
> - **VQ-VAE**: 300k parameters
> - **GPT-2**: 124M parameters
> - **Graphormer**: 50M parameters
>
> We have updated this information in our revised manuscript.

---

> > ### Comment · Reviewer_gMZe · 2024-11-26
> >
> > Thanks for the author’s rebuttal. Some of my concerns are not well addressed, and hope the authors can provide more explanation or details.
> >
> > 1. About my W1: The authors only talk about the difference with FoldSeek, what is the difference with FoldToken? In addition, I am not sure whether the differences (e.g. distinct features, training objective) are indeed important. It would be better to provide additional ablation study. Also, I don't think the training objective is a novel point since the used training objective is the same as VQ-VAE.
> >
> > 2. About my W4: about some visualization, I understand that it might not be easy to plot sth similar to FoldSeek, but I think additional visualization and analysis of the alphabet is still needed. Can the authors provide a similar plot as Figure 4 (a) on the angles?
> >
> > 3. In addition, I have a new question about the advantage of using VQ-VAE, specifically, the advantage of VQ. For example, how about first train an AE, and then use methods like k-means/clustering to assign to codebooks, as in this recent paper [1].
> >
> > Thanks.
> >
> > [1] ProSST: Protein Language Modeling with Quantized Structure and Disentangled Attention

---

> > > ### Author Response · Authors · 2024-11-27
> > > **Follow-up Response to Reviewer gMZe (1/2)**
> > >
> > > Dear Reviewer gMZe,
> > >
> > > Thank you for your reply. We would like to address your remaining concerns as follows. Since your feedback emphasizes generation capabilities, we focus this response primarily on analyzing **generation descriptors**, even though our experiments involve both understanding and generation descriptors.
> > >
> > > > **Q1: More discussion on FoldToken, the need for ablations, and perceived limitations in novelty**
> > > >
> > >
> > > **Comparison with FoldToken**
> > >
> > > We did not extensively discuss FoldToken because the modeling approaches are fundamentally different. Our method adopts an **atom-level VAE**, where each atom's coordinates are discretized independently and decoupled from the atom type. In contrast, FoldToken uses a **protein-level modeling approach**, jointly discretizing amino acid types and structures, requiring a Transformer-based encoder-decoder architecture.
> > >
> > > Our motivation for adopting a purely MLP-based atom-level auto-encoder architecture is twofold:
> > >
> > > 1. **Task Complexity**: Atom-level modeling achieves excellent RMSD performance for molecular structures, making it sufficient for our target tasks.
> > > 2. **Generalizability**: By using an atom-level VAE, the resulting discrete structural vocabulary can generalize across different atomic types in various molecules rather than being tailored to specific atom type. To clarify, we have now included additional heatmap visualizations in the Appendix D to demonstrate this generalization.
> > >
> > > We also attempted a molecule-level Transformer encoder-decoder architecture similar to FoldToken, optimizing atom type loss, VQ loss, reconstruction loss, and sign loss. However, even with a vocabulary size of 8192, we could not achieve comparable performance due to time limitations. Furthermore, given the **efficiency and simplicity** of the atom-level VAE, we ultimately chose this approach. Below, we include a performance comparison:
> > >
> > > |  | **Atom Accuracy** | **Sign Accuracy** | **Gen-Length** | **Gen-Polar** | **Gen-Azimuth** |
> > > | --- | --- | --- | --- | --- | --- |
> > > | Molecule-Level | 97.9% | 98.4% | 0.07 | 0.55 | 0.26 |
> > > | Atom-Level (Ours) | - | 100% | 0.01 | 0.06 | 0.06 |
> > >
> > > **Ablation Results**
> > >
> > > We sincerely thank the reviewer for suggesting additional ablation studies. Below, we provide results for several ablations. Key ablations have also been added to our revised manuscript.
> > >
> > > |  | **Gen-Length** | **Gen-Polar** | **Gen-Azimuth** |
> > > | --- | --- | --- | --- |
> > > | Mol-StrucTok | 0.01 | 0.06 | 0.06 |
> > > | - w.o. Feature Normalization | 0.03 | 0.12 | 0.11 |
> > > | - w.o. Sign Prediction | 0.01 | 0.08 | 0.84 |
> > > | GNLL (Gaussian NLL) Instead of MSE | 0.22 | 0.54 | 0.46 |
> > >
> > > **Clarifying Novelty**
> > >
> > > We would like to clarify that we do not claim differences in training objectives as our novelty but rather to distinguish our approach from FoldSeek. While we understand novelty can be subjectively assessed, we reiterate our contributions to provide a comprehensive perspective as following:
> > >
> > > 1. **3D Line Notation for Molecules**: We propose a **specialized 3D line notation** for molecules, which is general and applicable to existing line notations. It uniquely represents a molecule’s 2D graph and allows reconstruction of the 3D structure based on the 2D graph.
> > > 2. **Discrete Mapping for Language Models**: To enable language model-based modeling—a relatively underexplored direction—we follow FoldSeek’s strategy of learning a discrete mapping over continuous descriptors.
> > > 3. **General-Purpose 3D Tokenizer**: By encoding each atom’s local neighborhood information, our tokenizer produces a vocabulary that not only uniquely reconstructs a molecule's 3D structure but also provides effective 3D representations.
> > > 4. **Performance Demonstration**: Through experiments with GPT-based modeling, we show that Mol-StrucTok performs well in molecular generation tasks. Furthermore, Graphormer-based evaluations demonstrate that Mol-StrucTok provides meaningful 3D information for molecular understanding tasks.
> > >
> > > We hope this response addresses your concerns and provides a clearer understanding of our work’s contributions and decisions.

---

> > > ### Author Response · Authors · 2024-11-27
> > > **Follow-up Response to Reviewer gMZe (2/2)**
> > >
> > > > **Q2: More visualization analysis**
> > > >
> > >
> > > Thank you for your constructive feedback. We agree that additional visualization and analysis can indeed further illustrate the significance of Mol-StrucTok.
> > >
> > > **1. Decoding on Angular Descriptors**
> > >
> > > In addition to bond distances, we also provide the angular descriptors distribution visualizations, including polar and azimuthal angles. The results are shown in Figure 5 in the appendix, where we decoded the 256-token vocabulary with respect to the polar and azimuthal angles.
> > >
> > > Although the specific meanings of individual tokens in the vocabulary are not directly interpretable, the distributions clearly exhibit distinct peaks for both polar and azimuthal angles. This indicates that the angular representations derived from our **local spherical coordinates** are well-concentrated. Furthermore, our VQVAE adaptively learns this concentration by allocating more tokens to regions with higher density, which supports the effectiveness of our tokenization strategy.
> > >
> > > **2. Atom Type and Shared Alphabets**
> > >
> > > Besides angular descriptors, we provide additional useful visualization results in Figure 6 in the appendix. A key feature of our VQVAE design is the **decoupling of atom types and local structural environments**. This enables certain tokens in the vocabulary to represent shared local environments across all atom types, while others are specific to individual atom types. This design promotes a **compact and generalized representation** of molecular structures.
> > >
> > > To evaluate this property, we analyzed the distribution of different atom types across token IDs in the PCQM4Mv2 dataset. The heatmap demonstrates that:
> > >
> > > - The vocabulary is highly utilized, with nearly all token IDs assigned to meaningful local structures.
> > > - Most token IDs represent shared local structures across multiple atom types, showcasing the generalization capability of our approach.
> > > - A few token IDs, such as 19, 30, and 33, are specific to the local environments of particular atom types, indicating that our design also captures structural specificity when necessary.
> > >
> > > This analysis highlights that our vocabulary effectively balances **compactness and specificity** in structural representation.
> > >
> > > > **Q3: AE+clustering**
> > > >
> > >
> > > Thank you for raising this insightful question. It is a valuable suggestion, as VQ-VAE can be interpreted as performing clustering to provide compact discrete representations. Similarly, methods like ProSST, which directly perform clustering in the embedding space, can also yield compact discrete representations, aligning with the approach of AE + clustering that you mentioned.
> > >
> > > To further investigate this, we conducted additional experiments to compare VQ-VAE with AE followed by k-means clustering (with 256 clusters, the same size as our VQ-VAE vocabulary). As expected, AE achieves significantly better reconstruction performance compared to VQ-VAE. However, when applying k-means clustering on the AE’s embedding space, we observe slightly inferior results compared to VQ-VAE.
> > >
> > > The following table summarizes the results:
> > >
> > > | **Method** | **Gen-Length** | **Gen-Polar** | **Gen-Azimuth** |
> > > | --- | --- | --- | --- |
> > > | **VQ-VAE** | 0.012 | 0.064 | 0.064 |
> > > | **AE** | 0.009 | 0.022 | 0.036 |
> > > | **AE + Clustering** | 0.013 | 0.086 | 0.082 |
> > >
> > > These results highlight an interesting finding: while AE provides better reconstruction performance, the clustering step introduces some loss of precision, leading to a slightly inferior performance in generation tasks compared to VQ-VAE. This reinforces the advantage of VQ-VAE in learning representations that are inherently optimized for discrete codebooks during training, rather than relying on post-hoc clustering.
> > >
> > > We have included these findings and discussions in the revised manuscript to provide a more comprehensive analysis. Thank you again for the valuable suggestion—it has been helpful in strengthening the discussion and insights of our work.
> > >
> > > ---
> > >
> > > Thank you for taking the time to review our work. We hope this response sufficiently addresses your concerns and highlights the value and contributions of our research. Since the discussion period is approaching the end, we would greatly appreciate your reconsideration of the score for our submission.
> > >
> > > Sincerely,
> > >
> > > Authors

---

> ### Author Response · Authors · 2024-12-01
> **Further discussion request**
>
> Dear Reviewer gMZe,
>
> Thank you for your thoughtful feedback on our manuscript. We have carefully addressed all of the concerns you raised in our responses, and we hope that the clarifications provided sufficiently resolve the points of uncertainty and potentially alter any negative impressions that may have arisen during the initial review process. Given that the discussion period is nearing its deadline, we kindly request that you review our responses at your earliest convenience.
>
> We greatly appreciate your time and effort in reviewing our work and look forward to your feedback.
>
> Best regards,
>
> Authors

---

> > ### Comment · Reviewer_gMZe · 2024-12-02
> >
> > Thank you for the authors' response. Most of my concerns are well addressed, and I increased my score to 5. Overall, I think this is an interesting paper, but with some similarities to existing methods. Also please consider including larger molecule datasets (such as GEOM-Drug) in the next version. Thanks again.

---

> ### Author Response · Authors · 2024-12-02
> **Response to Reviewer gMZe**
>
> Dear Reviewer gMZe,
>
> We appreciate your feedback for acknowledging the improvements we've made in addressing your concerns. **Regarding the observation that our work appears similar to previous studies, we would like to emphasize that both works were developed concurrently.** The similarity you noted only reinforces the idea that this is an actively explored topic. **In our paper and rebuttals, we have provided detailed explanations of how our approach differs from those previously proposed.** As the Area Chair has not provided explicit feedback, we understand that we must accept your judgment regarding this matter.
>
> Once again, thank you for your time and consideration. Wishing you all the best.
>
> Kind regards,
>
> Authors

---

### Author Response · Authors · 2024-11-25

Dear Reviewers,

Thanks for the time, effort, and expertise you have invested in reviewing our paper.

As the author-reviewer discussion period approaches its deadline, we would greatly appreciate it if you could prioritize providing your feedback at your earliest convenience.

Thank you again for your valuable suggestions and contributions.


Sincerely,

Authors

---

### Author Response · Authors · 2024-11-28
**Official Comment Regarding Reviewer Violation of ICLR's Contemporaneous Work Policy**

Dear Area Chairs, Senior Area Chairs, and Program Chairs,

We deeply appreciate the review process and value the constructive feedback we have received. However, we would like to bring to your attention a concern regarding Reviewer **8GHQ,** whose evaluation appears to contravene the ICLR policy on contemporaneous work comparisons.

According to the ICLR Reviewer Guide (https://iclr.cc/Conferences/2025/ReviewerGuide), authors are not expected to compare their work to papers published within four months before the submission deadline or to non-peer-reviewed works (e.g., papers on arXiv). Specifically, the policy states:

> We consider papers contemporaneous if they are published within the last four months. That means, since our full paper deadline is October 1, if a paper was published (i.e., at a peer-reviewed venue) on or after July 1, 2024, authors are not required to compare their own work to that paper.
>

The reviewer **has explicitly based his/her rejection recommendation on a direct comparison with a contemporaneous work posted on arXiv (not a publishment) on August 19, 2024**, citing the absence of one additional dataset and slightly lower performance on a few metrics as reasons. However, even at the time of submission, the overall performance of our work was comparable to the contemporaneous study (listed below). Furthermore, our experiments on QM9 were carefully designed to be representative of the task, in alignment with prior ICLR papers such as **Symphony** (https://openreview.net/forum?id=MIEnYtlGyv), which also evaluated methods solely on QM9.

| Unconditional | Atom Sta | Mol Sta | Valid | Valid & Unique |
| --- | --- | --- | --- | --- |
| Geo2Seq | 98.3% | **90.3%** | 94.8% | 80.6% |
| Mol-StrucTok | **98.5%** | 88.3% | **98.0%** | **85.1%** |

| Conditional | alpha | gap | homo | lumo | mu | Cv |
| --- | --- | --- | --- | --- | --- | --- |
| Geo2Seq | 0.46 | 98 | **57** | 71 | **0.164** | 0.275 |
| Mol-StrucTok | **0.33** | **89** | 64 | **62** | 0.285 | **0.169** |

While we respect the value of the contemporaneous work and acknowledge its inspiring contributions, **we believe it is inappropriate for this to form the basis of a rejection recommendation given that it directly conflicts with ICLR’s stated policy, and also given the results of both papers are comparable.** Our submission included a detailed discussion highlighting the distinctions between our work and the referenced study, demonstrating that both contributions are meaningful to the field.

We kindly request your guidance and consideration on this matter, as the evaluation may have been unduly influenced by factors outside the scope of ICLR’s policies. We remain open to constructive feedback and are committed to addressing valid concerns to improve our work further.

Thank you for your time and attention to this matter.

Sincerely,

Authors

---

> ### Author Response · Authors · 2024-12-02
> **Kindly request for review-author discussion help**
>
> Dear AC and SAC,
>
> We truly appreciate the reviewing process and the efforts paid by reviewers.
> With only one day remaining until the discussion period ends, **we kindly request your help in helping with the author-review discussion**. We have paid much efforts to add more experiments and tried our best to address the concerns raised by reviewers. **Even as the experiments compared to the paper uploaded on Arxiv on Aug, which is not necessarily needed that avoids the policy**. However, it is sad that the reviewers do not give much attention on our rebuttals and follow-up responses.
>
> **We do concern that we have received some biased reviews and we sincerely hope you could help to guide the discussion period and also  give a fair evaluation on our paper, instead of only considering the reviewers' initial comments. We do look forward your kind help.**
>
> Appreciate a lot.
>
> Authors.

---

### Meta-Review · Area_Chair_RE8L · 2024-12-22

**Metareview:**

This paper studies how to convert molecules into tokens for language model generation. The overall idea is quite interesting, but the reviewers raised major concerns on the technical and experimental aspects of the work, which are not resolved after rebuttals. I agree that this work does not necessarily need to compare with Geo2Seq due to its relative timing. However, the reviewers are concerned with method novelty and experimental advantages compared to other prior methods, like FoldSeek, FoldToken, VQ-VAE, etc. in terms of general methods, loss functions, and empirical improvements. Due to these major unresolved concerns, a reject is recommended.

**Additional Comments On Reviewer Discussion:**

Extensive discussions have been involved and some of the major issues are still not resolved, including comparison with multiple prior methods experimentally and conceptually.

---

### Decision · Program_Chairs · 2025-01-22

Reject